# Determination of host proteins composing the microenvironment of coronavirus replicase complexes by proximity-labeling

Philip V'kovski[1,2,3], Markus Gerber[1,2], Jenna Kelly[1,2,4], Stephanie Pfaender[1,2], Nadine Ebert[1,2], Sophie Braga Lagache[5], Cedric Simillion[5,6], Jasmine Portmann[1,2], Hanspeter Stalder[1,2], Véronique Gaschen[7], Rémy Bruggmann[4], Michael H Stoffel[7], Manfred Heller[5], Ronald Dijkman[1,2], Volker Thiel[1,2]*

[1]Institute of Virology and Immunology IVI, Bern, Switzerland; [2]Department of Infectious Diseases and Pathobiology, Vetsuisse Faculty, University of Bern, Bern, Switzerland; [3]Graduate School for Biomedical Science, University of Bern, Bern, Switzerland; [4]Interfaculty Bioinformatics Unit, SIB Swiss Institute of Bioinformatics, University of Bern, Bern, Switzerland; [5]Mass Spectrometry and Proteomics Core Facility, Department for BioMedical Research (DBMR), University of Bern, Bern, Switzerland; [6]Department of Clinical Research, University of Bern, Bern, Switzerland; [7]Division of Veterinary Anatomy, Vetsuisse Faculty, University of Bern, Bern, Switzerland

**Abstract** Positive-sense RNA viruses hijack intracellular membranes that provide niches for viral RNA synthesis and a platform for interactions with host proteins. However, little is known about host factors at the interface between replicase complexes and the host cytoplasm. We engineered a biotin ligase into a coronaviral replication/transcription complex (RTC) and identified >500 host proteins constituting the RTC microenvironment. siRNA-silencing of each RTC-proximal host factor demonstrated importance of vesicular trafficking pathways, ubiquitin-dependent and autophagy-related processes, and translation initiation factors. Notably, detection of translation initiation factors at the RTC was instrumental to visualize and demonstrate active translation proximal to replication complexes of several coronaviruses. Collectively, we establish a spatial link between viral RNA synthesis and diverse host factors of unprecedented breadth. Our data may serve as a paradigm for other positive-strand RNA viruses and provide a starting point for a comprehensive analysis of critical virus-host interactions that represent targets for therapeutic intervention.
DOI: https://doi.org/10.7554/eLife.42037.001

*For correspondence:
volker.thiel@vetsuisse.unibe.ch

Competing interests: The authors declare that no competing interests exist.

## Introduction

Positive-strand RNA viruses replicate at membranous structures that accommodate the viral replication complex and facilitate RNA synthesis in the cytosol of infected host cells (*Romero-Brey and Bartenschlager, 2016*; *Romero-Brey et al., 2012*; *Cortese et al., 2017*; *Knoops et al., 2008*; *Miorin et al., 2013*). Rewiring host endomembranes is hypothesized to provide a privileged microenvironment physically separated from the cytosol, thereby ensuring adequate concentrations of macromolecules for viral RNA synthesis, preventing recognition of replication intermediates such as double-stranded RNA (dsRNA) by cytosolic innate immune receptors (*Overby et al., 2010*;

**eLife digest** Coronaviruses can infect the nose and throat and are a main cause of the common cold. Infections are usually mild and short-lived, but sometimes they can turn nasty. In 2002 and 2012, two dangerous new coronaviruses emerged and caused diseases known as SARS and MERS. These viruses caused much more serious symptoms and in some cases proved deadly. The question is, why are some coronaviruses more dangerous than others? Scientists know that the body's response to virus infection can make a difference to whether someone had mild or severe disease. So, to understand why some coronaviruses cause a cold and others kill, they also need to learn how people react to virus infection.

Coronaviruses hijack membranes inside cells and turn them into virus factories. Within these factories, the viruses build molecular machinery called replicase complexes to copy their genetic code, which is needed for the next generation of virus particles. The viruses steal and repurpose proteins from their host cell that will assist in the copying process. However, scientists do not yet know which host proteins are essential for the virus to multiply. So, to find out, V'kovski et al. developed a way to tag any host protein that came near the virus factories.

The new technique involved attaching an enzyme called a biotin ligase to the replicase complex. This enzyme acts as a molecular label gun, attaching a chemical tag to any protein that comes within ten nanometres. The label gun revealed that more than 500 different proteins come into contact with the replicase complex. To find out what these proteins were doing, the next step was to switch off their genes one by one. This revealed the key cell machinery that coronaviruses hijack when they are replicating. It included the cell's cargo transport system, the waste disposal system, and the protein production system. Using these systems allows the viruses to copy their genetic code next to machines that can turn it straight into viral proteins.

These new results provide clues about which proteins viruses actually need from their host cells. They also do not just apply to coronaviruses. Other viruses use similar strategies to complete their infection cycle. These findings could help researchers to understand more generally about how viruses multiply. In the future, this knowledge could lead to new ways to combat virus infections.
DOI: https://doi.org/10.7554/eLife.42037.002

*Neufeldt et al., 2016*), and providing a platform that facilitates molecular interactions with host cell proteins.

Ultrastructural studies have reported the origin, nature, and extent of membrane modifications induced by coronaviruses (order *Nidovirales*, family *Coronaviridae*), which materialize as an ER-derived network of interconnected double-membrane vesicles (DMVs) and convoluted membranes (CM) in perinuclear regions of infected cells to which the viral replication/transcription complex (RTC) is anchored (*Knoops et al., 2008*; *Ulasli et al., 2010*; *Oudshoorn et al., 2017*). The RTC is generated by translation of the genomic RNA into two large polyproteins that are extensively auto-proteolytically processed by viral proteases to give rise to 16 processing end-products, termed non-structural proteins (nsps) 1–16. Nsp1 is rapidly cleaved from the polyproteins and not considered an integral component of the coronaviral RTC, but interferes with host cell translation by inducing degradation of cellular mRNAs (*Huang et al., 2011*; *Züst et al., 2007*; *Lokugamage et al., 2015*). Although it has not yet been formally demonstrated, the remaining nsps (nsp2-16) are thought to comprise the RTC and harbor multiple enzymes and functions, such as de-ubiquitination, proteases, helicase, polymerase, exo- and endonuclease, and N7- and 2'O-methyltransferases (*Thiel et al., 2003*; *Snijder et al., 2003*; *Decroly et al., 2008*; *Barretto et al., 2005*; *Lindner et al., 2005*; *Athmer et al., 2017*). Many of these functions have been studied using reverse genetic approaches, which revealed their importance in virus-host interactions (*Kindler et al., 2017*; *Züst et al., 2011*; *Eckerle et al., 2007*; *Deng et al., 2017*; *Zhang et al., 2015*). In most cases, phenotypes were described via loss-of-function mutagenesis. However, in the context of virus infection, the specific interactions of RTC components with host cell factors remain largely unknown.

A number of individual host cell proteins have been shown to impact coronavirus replication by using various screening methods, such as genome-wide siRNA, kinome, and yeast-two-hybrid screens (*Verheije et al., 2008*; *Reggiori et al., 2010*; *de Wilde et al., 2015*; *Wong et al., 2015*;

*Pfefferle et al., 2011*). Likewise, genome-wide CRISPR-based screens have been applied to other positive-stranded RNA viruses, such as flaviviruses, and identified critical host proteins required for replication (*Marceau et al., 2016*; *Zhang et al., 2016*). Some of these proteins were described in the context of distinct ER processes, such as N-linked glycosylation, ER-associated protein degradation (ERAD), and signal peptide insertion and processing. Although individual proteins identified by these screens may interact with viral replication complexes, they likely constitute only a small fraction of the global replicase microenvironment.

To capture the full breadth of host cell proteins and cellular pathways that are spatially associated with viral RTCs, we employed a proximity-based labeling approach involving a promiscuous *E. coli*-derived biotin ligase (BirA$_{R118G}$). BirA$_{R118G}$ biotinylates proximal (<10 nm) proteins in live cells without disrupting intracellular membranes or protein complexes, and hence, does not rely on high-affinity protein-protein interactions but is also able to permanently tag transient interactions (*Roux et al., 2012*). Covalent protein biotinylation allows stringent lysis and washing conditions during affinity purification and subsequent mass spectrometric identification of captured factors. By engineering a recombinant MHV harboring BirA$_{R118G}$ as an integral component of the RTC, we identified >500 host proteins reflecting the molecular microenvironment of MHV replication structures. siRNA-mediated silencing of each of these factors highlighted, amongst others, the functional importance of vesicular ER-Golgi apparatus trafficking pathways, ubiquitin-dependent and autophagy-related catabolic processes, and translation initiation factors. Importantly, the detection of active translation in close proximity to the viral RTC highlighted the critical involvement of translation initiation factors during coronavirus replication. Collectively, the determination of the coronavirus RTC-associated microenvironment provides a functional and spatial link between conserved host cell processes and viral RNA synthesis, and highlights potential targets for the development of novel antiviral agents.

## Results

### Engineering the BirA$_{R118G}$ biotin ligase into the MHV replicase transcriptase complex

To insert the promiscuous biotin ligase BirA$_{R118G}$ as an integral subunit of the MHV RTC, we used a vaccinia virus-based reverse genetic system (*Coley et al., 2005*; *Eriksson et al., 2008*) to generate a recombinant MHV harboring an in-frame fusion of myc-tagged BirA$_{R118G}$ to nsp2. MHV-BirA$_{R118G}$-nsp2 retained the cleavage site between nsp1 and BirA$_{R118G}$, while a deleted cleavage site between BirA$_{R118G}$ and nsp2 ensured the expression of a BirA$_{R118G}$-nsp2 fusion protein (*Figure 1a*). This strategy was chosen because it was recently employed by Freeman *et al.* for a fusion of green fluorescent protein (GFP) with nsp2 and represents the only known site tolerating large insertions within the MHV replicase polyprotein (*Freeman et al., 2014*). MHV-BirA$_{R118G}$-nsp2 replicated to comparable peak titers and replication kinetics as the parental wild-type MHV-A59 (*Figure 1b*). MHV-GFP-nsp2, which was constructed in parallel and contained the coding sequence of EGFP (*Freeman et al., 2014*) instead of BirA$_{R118G}$, was used as a control and also reached wild-type virus peak titers, with slightly reduced viral titers at 9 hr post- infection (h.p.i.) compared to MHV-A59 and MHV-BirA$_{R118G}$-nsp2 (*Figure 1b*).

Western blot analysis confirmed that the BirA$_{R118G}$-nsp2 fusion protein is specifically detected in MHV-BirA$_{R118G}$-nsp2-infected cells and that the BirA$_{R118G}$ biotin ligase remains fused to nsp2 during MHV-BirA$_{R118G}$-nsp2 infection (*Figure 1—figure supplement 1*). To further confirm the accommodation of BirA$_{R118G}$ within the viral RTC, MHV-A59-, MHV-BirA$_{R118G}$-nsp2-, and mock-infected L929 fibroblasts were visualized using indirect immunofluorescence microscopy. BirA$_{R118G}$-nsp2 remained strongly associated with the MHV RTC throughout the entire replication cycle, as indicated by the co-localization of BirA$_{R118G}$-nsp2 with established markers of the MHV replicase, such as nsp2/3 and nsp8 (*Figure 1c*, *Figure 1—figure supplement 2*, *Figure 1—figure supplement 3*). This observation corroborates previous studies demonstrating that nsp2, although not required for viral RNA synthesis, co-localizes with other nsps of the coronaviral RTC (*Schiller et al., 1998*; *Hagemeijer et al., 2010*; *Graham et al., 2005*). Importantly, by supplementing the culture medium with biotin, we could readily detect biotinylated proteins with fluorophore-coupled streptavidin that appeared close to the MHV RTC throughout the entire replication cycle in MHV-BirA$_{R118G}$-nsp2-infected cells,

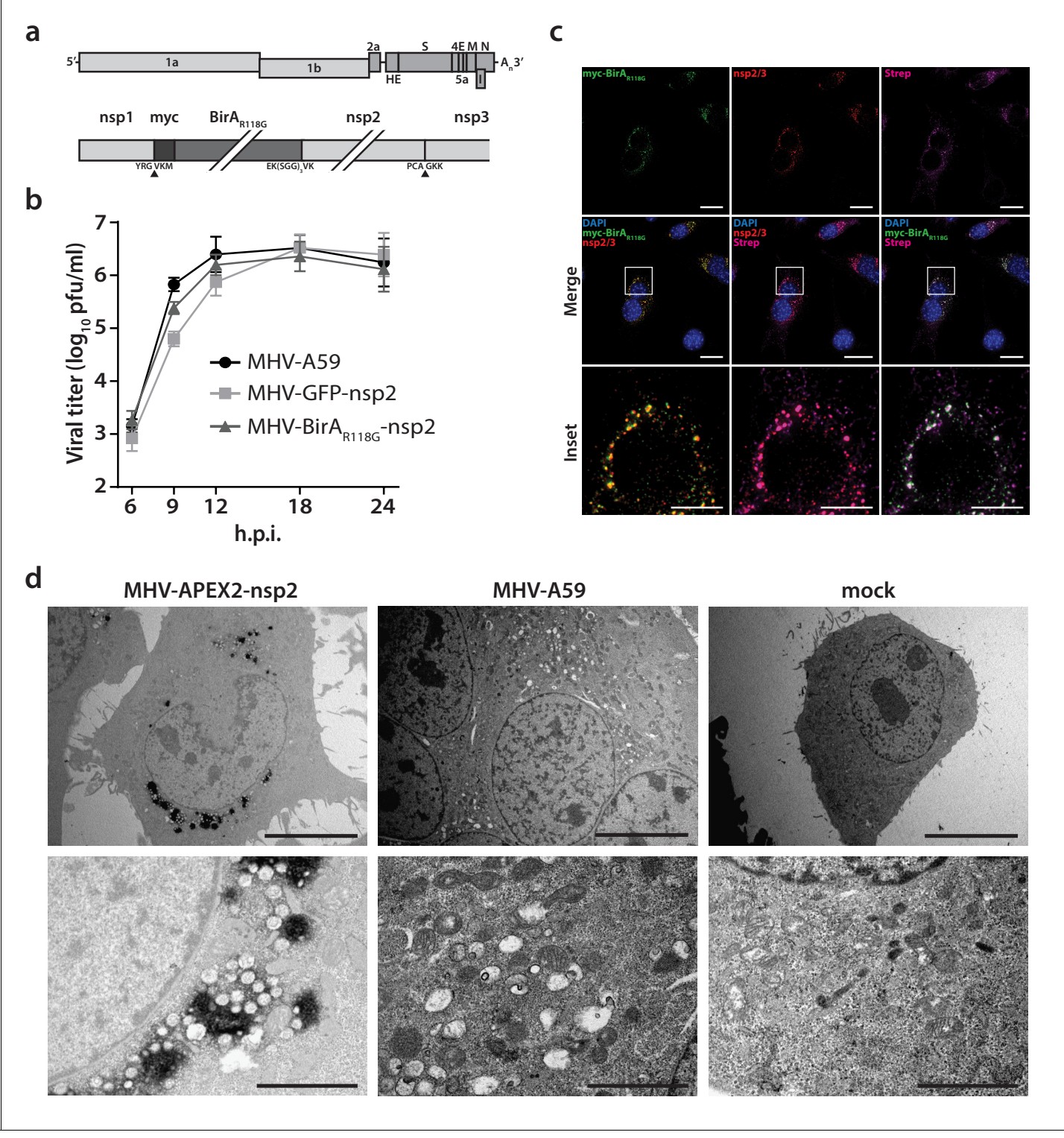

**Figure 1.** Characterization of the recombinant MHV-BirA(R118G)-nsp2. (**a**) Genome organization of recombinant MHV-BirA(R118G)-nsp2. The positive-sense RNA genome of MHV contains a 5' cap and a 3' poly(A) tail. ORF1a and ORF1b encode the viral replication and transcription complex (nsp1-16). myc-BirA(R118G) was inserted as an N-terminal fusion with nsp2 within ORF1a. The cleavage site between nsp1 and myc-BirA(R118G) was retained (black arrow) while a deleted cleavage site between BirA(R118G) and nsp2 ensured the release of a BirA(R118G)-nsp2 fusion protein from the pp1a polyprotein. The cleavage site between nsp2 and nsp3 was also retained (grey arrow). (**b**) Viral replication kinetics of recombinant MHV-BirA(R118G)-nsp2 were compared to wild-type MHV-A59 and recombinant MHV-GFP-nsp2. Murine L929 fibroblasts were infected at a multiplicity of infection (MOI) of 1 plaque forming unit (pfu) per cell. Viral supernatants were collected at the indicated time points, titrated by plaque assay and expressed in pfu per ml. Data points

*Figure 1 continued on next page*

*Figure 1 continued*

represent the mean and SEM of three independent experiments, each performed in quadruplicate. (**c**) Immunofluorescence analysis of MHV-BirA$_{R118G}$-nsp2-mediated biotinylation of RTC-proximal factors. L929 cells were infected with MHV-BirA$_{R118G}$-nsp2 (MOI = 1) in medium supplemented with 67 μM biotin. Cells were fixed 15 hr post infection (h.p.i.) and processed for immunofluorescence analysis with antibodies directed against the BirA$_{R118G}$ (anti-myc), the viral replicase (anti-nsp2/3) and biotinylated factors (streptavidin). Nuclei are counterstained with DAPI. Z-projection of deconvolved z-stacks acquired with a DeltaVision Elite High-Resolution imaging system are shown. Scale bars: 20 μm; insets 5 μm. (**d**) Ultrastructural analysis of MHV-APEX2-nsp2 infection. L929 cells were infected with MHV-APEX2-nsp2 and MHV-A59 (MOI = 2), or mock infected. At 10 h.p.i., cells were fixed, stained with DAB and processed for electron microscopy investigations. Representative low (scale bar: 10 μm) and high magnifications (scale bar: 2 μm) are displayed.

DOI: https://doi.org/10.7554/eLife.42037.003

The following figure supplements are available for figure 1:

**Figure supplement 1.** Western blot detection of BirA$_{R118G}$-nsp2.
DOI: https://doi.org/10.7554/eLife.42037.004
**Figure supplement 2.** Immunofluorescence analysis of MHV-BirA$_{R118G}$-nsp2-mediated biotinylation.
DOI: https://doi.org/10.7554/eLife.42037.005
**Figure supplement 3.** Immunofluorescence analysis of the MHV-BirA$_{R118G}$-nsp2 RTC and BirA$_{R118G}$-nsp2 –mediated biotinylataion.
DOI: https://doi.org/10.7554/eLife.42037.006

demonstrating efficient proximity-dependent biotinylation of RTC-proximal host factors (*Figure 1c*, *Figure 1—figure supplement 2*, *Figure 1—figure supplement 3*).

Furthermore, to define the localization of the nsp2 fusion protein at the ultrastructural level, we replaced the BirA$_{R118G}$ biotin ligase with the APEX2 ascorbate peroxidase to generate recombinant MHV-APEX2-nsp2. APEX2 mediates the catalysis of 3,3'-diaminobenzidine (DAB) into an insoluble polymer that can be readily observed by electron microscopy (*Martell et al., 2017*). As shown in *Figure 1d*, APEX2-catalized DAB polymer deposition was readily detectable at characteristic corona-virus replication compartments, such as DMVs and CM, categorically demonstrating that the nsp2 fusion proteins localize to known sites of coronavirus replication (*Knoops et al., 2008*; *Ulasli et al., 2010*).

Collectively, these results establish that the recombinant MHV-BirA$_{R118G}$-nsp2 replicates with comparable kinetics to wild-type MHV-A59, expresses a functional BirA$_{R118G}$ biotin ligase that is tightly associated with the MHV RTC, and that biotinylated, RTC-proximal proteins can be readily detected in MHV-BirA$_{R118G}$-nsp2 infected cells.

## Determination of the coronavirus RTC-proximal proteome

To further demonstrate the efficiency and specificity of BirA$_{R118G}$-mediated biotinylation we assessed, by western blot analysis, fractions of biotinylated proteins derived from MHV-A59-, MHV-BirA$_{R118G}$-nsp2-, or non-infected cells that were grown with or without the addition of biotin (*Figure 2a*, *Figure 2b*). A characteristic pattern of endogenously biotinylated proteins was observed under all conditions where no exogenous biotin was added to the culture medium (*Figure 2b*). The same pattern was detectable in non-infected and wild-type MHV-A59-infected cells when the culture medium was supplemented with biotin, suggesting that the addition of biotin in the absence of the BirA$_{R118G}$ biotin ligase does not recognizably change the fraction of endogenously biotinylated pro-teins. In contrast, we observed a greatly increased fraction of biotinylated proteins in lysates derived from MHV-BirA$_{R118G}$-nsp2-infected cells treated with biotin. This result demonstrates that virus-mediated expression of the BirA$_{R118G}$ biotin ligase results in efficient biotinylation when biotin is added to the culture medium. Moreover, we could readily affinity purify, enrich, and recover the fraction of biotinylated proteins under stringent denaturing lysis and washing conditions by using streptavidin-coupled magnetic beads (*Figure 2b*).

Affinity purified proteins derived from biotin-treated MHV-A59- and MHV-BirA$_{R118G}$-nsp2-infected cells were subjected to mass spectrometric analysis (n = 3). Liquid chromatography tandem-mass spectrometry (LC-MS/MS) was performed from in-gel digested samples and log-transformed label-free quantification (LFQ) levels were used to compare protein enrichment between samples (*Figure 2c*). Overall, 1381 host proteins were identified, of which 513 were statistically significantly enriched in MHV-BirA$_{R118G}$-nsp2-infected samples over MHV-A59-infected samples. These host pro-teins represent a comprehensive repertoire of RTC-proximal factors throughout MHV infection

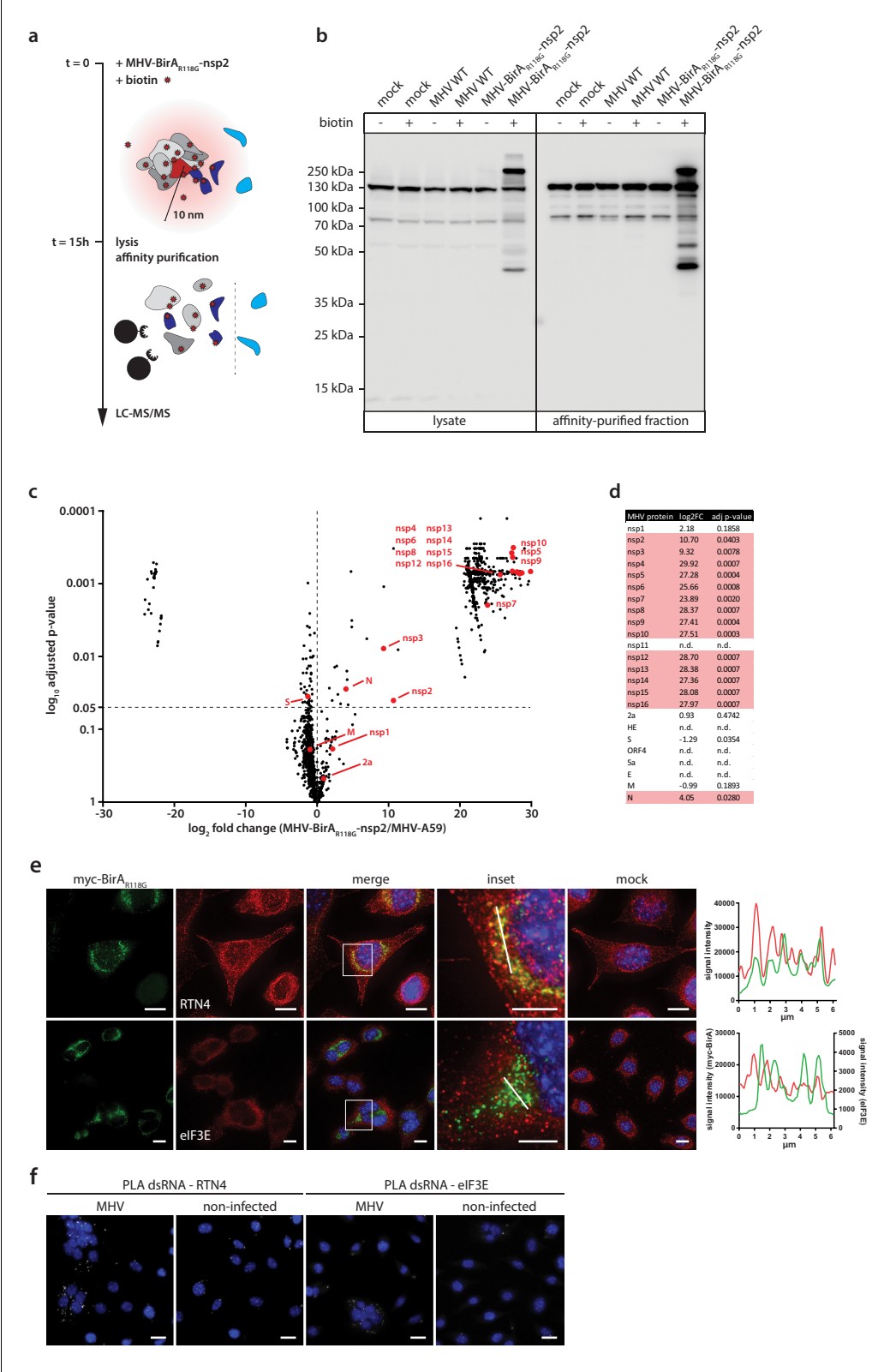

**Figure 2.** Determination of the coronavirus RTC-proximal proteome. (a) Schematic overview of the BirA$_{R118G}$-mediated proximity biotinylation assay using MHV-BirA$_{R118G}$-nsp2. (b) Western blot analysis of MHV-BirA$_{R118G}$-nsp2-infected L929 cells. L929 cells were infected with MHV-BirA$_{R118G}$-nsp2, MHV-A59 or non-infected in medium with and without supplementation of 67 μM biotin. Cells were lysed 15 h.p.i. and biotinylated factors were subjected to affinity purification using streptavidin-coupled magnetic beads. Total cell lysates and affinity-purified fractions were separated by SDS-

*Figure 2 continued on next page*

*Figure 2 continued*

PAGE and analysed by western blot probed with horse radish peroxidase (HRP)-coupled Streptavidin. (c) Host and viral factors identified by LC-MS/MS. $4*10^7$ L929 cells were infected with MHV-BirA$_{R118G}$-nsp2 or MHV-A59 in medium supplemented with 67 μM biotin. 15 h.p.i., lysates were affinity purified and LC-MS/MS was performed from in-gel digested samples. MS identification of biotinylated proteins was performed in three independent biological replicates. Spectral interpretation was performed against a Mus musculus and MHV database and log$_2$-transformed LFQ levels (x-axis) were used to determine significant differences in protein enrichment between sample groups (Student's T-test, y-axis). Identified cellular proteins are displayed as black dots, MHV proteins are highlighted in red (nsp: non-structural protein, N: nucleocapsid, S: spike, M: membrane, 2a: accessory protein 2a). (d) Summary of viral proteins identified by LC-MS/MS. nsp2-10, nsp12-16, and nucleocapsid were significantly enriched in fractions derived from MHV-BirA$_{R118G}$-nsp2-infected cells whereas nsp1, nsp11, structural proteins spike (S), envelope (E) and membrane proteins (M) as well as all accessory proteins (NS2a, HE, ORF4, ORF5a) were either not significantly enriched or not detected. (e,f) Immunofluorescence analysis of RTC-proximal cellular factors. L929 cells were seeded on coverslips, infected with MHV-BirA$_{R118G}$-nsp2 (e) or MHV-A59 (f), fixed at 9 h.p.i. and processed for immunofluorescence using anti-myc, anti-RTN4 and anti-eIF3E antibodies (e) or anti-dsRNA, anti-RTN4 and anti-eIF3E antibodies (f). Secondary fluorophore-coupled antibodies were used to detect the viral replicase and endogenous levels of RTN4 and eIF3E (e). Scale bars: 10 μm; insets 5 μm. Proximity ligations were performed using Duolink In Situ detection reagents (f). Nuclei are counterstained with DAPI. Z-projection of deconvolved z-stacks acquired with a DeltaVision Elite High-Resolution imaging system are shown. Intensity profiles highlighted in the magnified regions are shown. Scale bars: 20 μm (insets 5 μm).

DOI: https://doi.org/10.7554/eLife.42037.007

(*Figure 2c*, *Supplementary file 1*). Thirty-four factors, that are mainly involved in fatty acid β-oxidation biological processes in the mitochondrion, displayed significant enrichment in MHV-WT compared to MHV-BirA$_{R118G}$-nsp2 infections (*Figure 2c*). Since the biotinylation of these factors is not caused by the BirA$_{R118G}$ biotin ligase, these factors were not considered for further investigation in the context of our RTC-proximal biotinylation proteomic screen.

Importantly, besides the 513 host proteins that were enriched in MHV-BirA$_{R118G}$-nsp2-infected cells, we noted that viral replicase gene products nsp2-10 and nsp12-16, and the nucleocapsid protein were also significantly enriched in fractions derived from MHV-BirA$_{R118G}$-nsp2-infected cells, suggesting that they are all in close proximity to the BirA$_{R118G}$-nsp2 fusion protein (*Figure 2c*, *Figure 2d*). This is in agreement with studies demonstrating co-localization and interactions amongst individual nsps, and with studies showing association of the nucleocapsid protein with the coronavirus RTC (*Ulasli et al., 2010*; *Denison et al., 1999*; *Sims et al., 2000*; *van der Meer et al., 1999*; *Bost et al., 2001*). It also highlights the specificity and effectiveness of the labeling approach in live cells and is the first experimental evidence showing that collectively these viral nsps and the nucleocapsid (N) protein are subunits of the coronavirus RTC. Furthermore, these results corroborate previous reports that nsp1 is likely not an integral component of the coronavirus RTC (*Huang et al., 2011*; *Züst et al., 2007*; *Lokugamage et al., 2015*; *Denison et al., 1992*). Amongst the 'not detected' or 'not enriched' viral proteins are (i) nsp11, which is a short peptide of only 14 amino acids at the carboxyterminus of polyprotein 1a with a yet unassigned role or function in coronavirus replication, (ii) the structural proteins spike (S) protein, envelope (E) protein, and membrane (M) protein, which mainly localize to sites of viral assembly before being incorporated into newly-formed viral particles, and (iii) all accessory proteins (NS2a, HE, ORF4, ORF5a). Altogether, these results validate the proximity-dependent biotinylation approach and demonstrate the specific and exclusive labeling of MHV-RTC-associated proteins (*Figure 2d*).

The BirA$_{R118G}$ biotin ligase biotinylates proteins in its close proximity that must not necessarily have tight, prolonged, or direct interaction (*Roux et al., 2012*). Therefore, the identified RTC-proximal host proteins, recorded over the entire duration of the MHV replication cycle, likely include proteins that display a prolonged co-localization with the MHV RTC, proteins that may locate only transiently in close proximity to the RTC, and proteins of which only a minor fraction of the cellular pool may associate with the RTC. To this end, we assessed the localization of a limited number of host proteins from our candidate list in MHV-infected cells. Accordingly, we identified RTC-proximal host proteins displaying a pronounced co-localization with the MHV RTC, such as the ER protein reticulon 4 (RTN4; *Figure 2e*), and host proteins where co-localization by indirect immunofluorescence microscopy was not readily detectable, such as the eukaryotic translation initiation factor 3E (eIF3E; *Figure 2e*). However, in the latter case, a more sensitive detection technique, such as a proximity ligation assay that relies on proximity-dependent antibody-coupled DNA probe amplification

(*Söderberg et al., 2006*), demonstrated proximity of eIF3E and dsRNA in MHV-infected cells (*Figure 2f*).

Collectively, our results show that the approach of integrating a promiscuous biotin ligase as an integral subunit into a coronavirus RTC revealed a comprehensive list of host cell proteins that comprises the RTC microenvironment. The efficacy and specificity of our approach is best illustrated by the fact that we were able to identify all expected viral components of the MHV RTC, while other viral proteins, such as nsp1, structural proteins S, E, and M, and accessory proteins, were not amongst the significantly enriched proteins. Since the biotin-based proximity labeling was performed during the entire viral life cycle, our data likely also contains proteins that are only transiently present in the RTC microenvironment or only comprise a sub-fraction of the cellular pool in close proximity to the MHV RTC.

## Functional classification of RTC-proximal host factors

To categorize functionally related proteins from the list of RTC-proximal host proteins and identify enriched biological themes in the dataset, we performed a functional classification of RTC-proximal factors using Gene Ontology (GO) enrichment analysis. 86 GO biological process (BP) terms were significantly enriched in the dataset (p-value < 0.05), of which 32 terms were highly significant (p-value < 0.005) (*Figure 3a*, *Supplementary file 2*). Additional analysis using AmiGO revealed that 25 of these 32 highly significant GO BP terms fell into five broad functional categories, namely cell adhesion, transport, cell organization, translation, and catabolic processes. To examine these categories further, identify important cellular pathways within them, and extract known functional associations among RTC-proximal host proteins, we performed STRING network analysis on the RTC-proximal proteins in each category (*Figure 3b*, *Figure 3c*, *Figure 3—figure supplement 1*).

Despite 'cell-cell adhesion' scoring high, it likely represents a typical limitation of gene annotation databases, where many genes play multiple roles in numerous pathways and processes. Accordingly, most genes assigned to the GO BP term 'cell-cell adhesion' are also found in the other categories described below.

The category 'transport' included protein trafficking and vesicular-mediated transport pathways and comprised the majority of RTC-proximal factors (*Figure 3a*, *Figure 3b*). Protein interaction network analysis, using STRING, revealed at least four distinct clusters of interacting factors within this category (*Figure 3b*). Cluster I, protein transport, comprised nuclear transport receptors at nuclear pore complexes, such as importins and transportins. Interestingly, this cluster also contained Sec63, which is part of the Sec61 translocon (*Rapoport, 2007*) and has been implicated in protein translocation across ER membranes. The list of RTC-proximal factors also included signal recognition particles SRP54a and SRP68 proteins (*Supplementary file 2*) that promote the transfer of newly synthetized integral membrane proteins or secreted proteins across translocon complexes. Furthermore, the list contained Naca and BTF3, which prevent the translocation of non-secretory proteins toward the ER lumen (*Wiedmann et al., 1994*; *Gamerdinger et al., 2015*).

Cluster II included vesicle components, tethers and SNARE (Soluble N-ethylmaleimide-sensitive-factor Attachment protein Receptor) proteins characteristic of the COPII-mediated ER-to-Golgi apparatus anterograde vesicular transport pathway whereas, cluster III contained components of the COPI-related retrograde Golgi-to-ER transport machinery. Moreover, Cluster IV was comprised of proteins that mediate clathrin-coated vesicle (endosomal) transport between the plasma membrane and the trans-Golgi network (TGN), which is also closely associated with the actin cytoskeleton. Together with sorting nexins, cluster IV components can be regarded as regulating late-Golgi trafficking events and interacting with the endosomal system.

Many of the cellular processes and host proteins assigned to 'transport' (specifically in clusters II-IV) are also listed in the category 'cell organization' (*Figure 3a*, *Figure 3—figure supplement 1a*). However, this category actually extends the importance of vesicular transport as it also contains factors involved in the architecture, organization, and homeostasis of the ER and Golgi apparatus, and the cytoskeleton-supporting these organelles.

Notably, a number of MHV RTC-proximal factors were part of the host translation machinery and assigned to category 'translation' (*Figure 3a*, *Figure 3c*). We found enrichment of factors involved in translation initiation, particularly multiple subunits of eIF3 and eIF4 complexes, as well as eIF2, eIF5, the Ddx3y helicase, and the Elongation factor-like GTPase 1, which are required for the formation of 43S pre-initiation complexes, 48S initiation complexes, and the assembly of elongation-competent

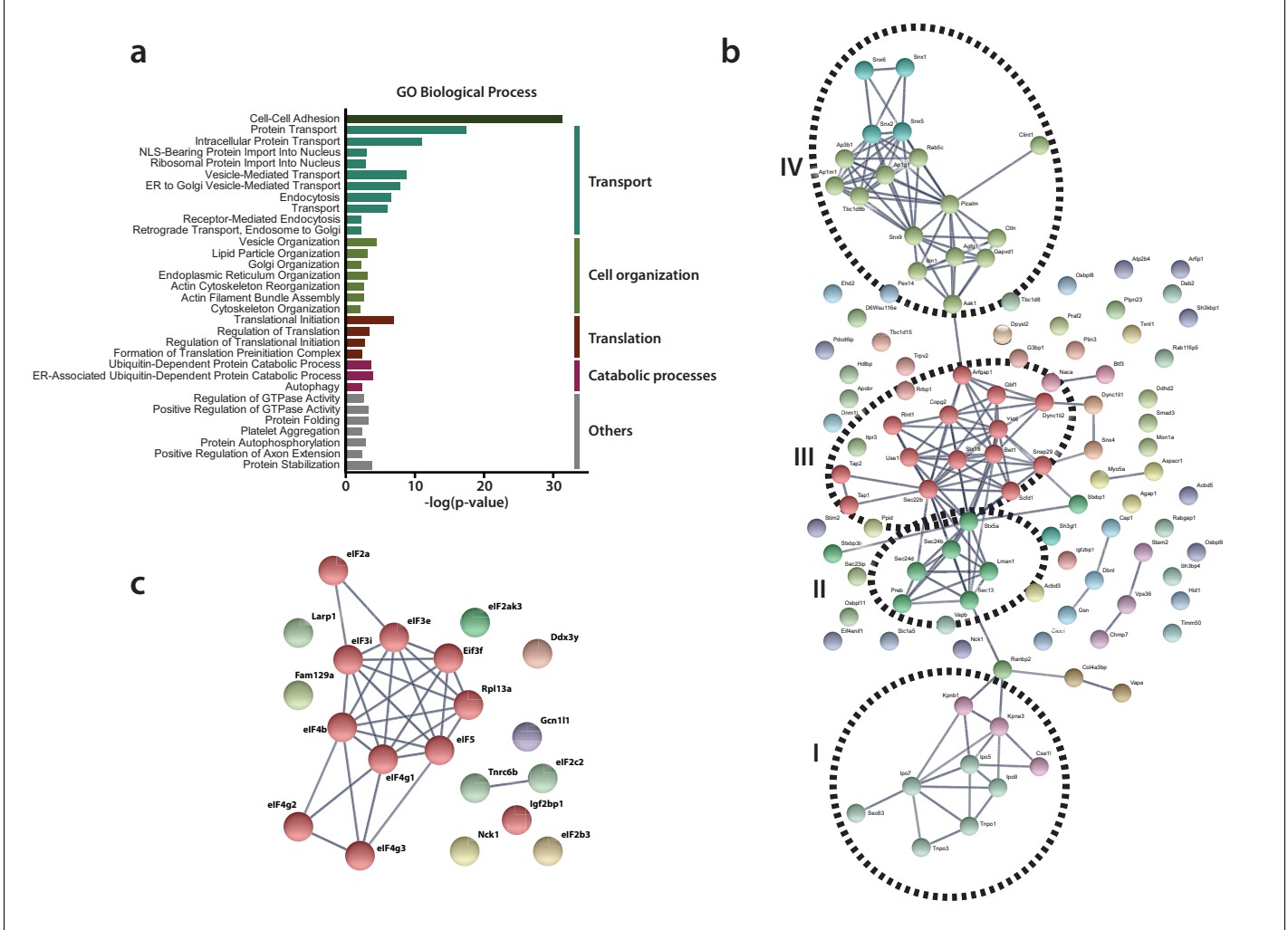

**Figure 3.** Functional classification of RTC-proximal host factors.  (a) Gene Ontology enrichment analysis of RTC-proximal cellular factors. 32 terms were highly significant (p-value < 0.005) and were assigned to five broad functional categories: cell-cell adhesion, transport, cell organization, translation, catabolic processes. (b–c) STRING protein interaction network analysis of the categories 'transport' (b) and 'translation' (c). The nodes represent RTC-proximal host proteins and the edges represent the interactions, either direct (physical) or indirect (functional), between two proteins in the network. Cellular proteins assigned to the 'transport' category separated into four distinct interaction clusters. I: protein transport, II: COPII anterograde transport, III: COPI retrograde transport, IV: clathrin-mediated transport.

DOI: https://doi.org/10.7554/eLife.42037.008

The following figure supplement is available for figure 3:

**Figure supplement 1.** STRING protein interaction network analysis of the categories 'cell organization' (a) and 'catabolic processes' (b).

DOI: https://doi.org/10.7554/eLife.42037.009

80S ribosomes (*Jackson et al., 2010*). The high degree of interaction between these subunits is suggestive of the presence of the entire translation initiation apparatus in close proximity to the viral RTC. The 60S ribosomal protein L13a (Rpl13a), ribosome biogenesis protein RLP24 (Rsl24d1), ribosome-binding protein 1 (Rbp1), release factor Gspt1, and regulatory elements, such as Igf2bp1, Gcn1l1, Larp, Fam129a and Nck1, are further indicative of the host cell translation machinery near sites of viral RNA synthesis.

Lastly, the category 'catabolic processes' (*Figure 3a*, *Figure 3—figure supplement 1b*) includes a subset of autophagy-related factors and numerous ubiquitin-dependent ERAD components, including the E3 ubiquitin-protein ligase complex and 26S proteasome regulatory subunits (Psmc2, Psmd4).

Collectively, the coronavirus RTC-proximal proteins identified by proximity labeling greatly expand the repertoire of candidate proteins implicated in the coronavirus replication cycle. Importantly, since this screening approach was tailored to detect host factors associated with the coronavirus RTC, it provides a spatial link of these factors to the site of viral RNA synthesis.

## Identification of proviral factors within the coronavirus RTC microenvironment

In order to assess the potential functional relevance of RTC-proximal factors identified in our MHV-BirA$_{R118G}$-nsp2-mediated proximity-dependent screen, we designed a custom siRNA library individually targeting the expression of each of the 513 identified RTC-proximal host proteins. siRNA-treated L929 cells were infected (MOI = 0.05, n = 4) with a recombinant MHV expressing a *Gaussia* luciferase reporter protein (MHV-Gluc) (*Lundin et al., 2014*) and replication was assessed by virus-mediated *Gaussia* luciferase expression (*Figure 4a*). Cell viability after siRNA knockdown was also assessed and genes resulting in cytotoxicity following silencing were discarded from further analysis. Importantly, we included internal controls of known relevance for MHV entry (MHV receptor Ceacam1a) and replication (Gbf1, Arf1) on each plate and found in each case that siRNA silencing of these factors significantly reduced MHV replication, which underscores the robustness and effectiveness of our approach (*Figure 4—figure supplement 1a*) (*Verheije et al., 2008*). We found that siRNA-mediated silencing of 53 RTC-proximal host factors significantly reduced MHV replication compared to non-targeting siRNA controls. These factors can therefore be considered proviral and required for efficient replication (*Figure 4b*; *Supplementary file 3*). In contrast, we did not find antiviral factors that resulted in significant enhancement of viral replication upon siRNA knockdown. While this work was performed in a murine fibroblast cell line, the identification of antiviral proteins may be anticipated in a similar siRNA-mediated knockdown screen using primary target cells such as macrophages, that are better equipped in eliciting antiviral responses upon virus infection.

Notably, siRNA targets that had the strongest impact on MHV replication were in majority contained within the functional categories highlighted in *Figure 3a* (*Figure 4b*). Indeed, in line with the hypothesis that MHV subverts key components mediating both anterograde and retrograde vesicular transport between the ER, Golgi apparatus and endosomal compartments for the establishment of replication organelles, several factors contained within these pathways impaired MHV replication as exemplified by the siRNA-mediated silencing of Kif11, Snx9, Dnm11, Scfd1, Ykt6, Stx5a, Clint1, Aak11, or Vapa (*Figure 4b*). Consistently, ER-associated protein sorting complexes associated with the ribosome and newly synthetized proteins (Naca, BTF3, SRP54a, SRP68) that were revealed in the GO enrichment analysis (*Figure 3a*, *Supplementary file 2*), also appear to be required for efficient MHV replication (*Figure 4b*).

Furthermore, we also observed significantly reduced MHV replication upon silencing of core elements of the 26S and 20S proteasome complex (Psmd1 and Psmc2, and Psmb3, respectively), suggesting a crucial role of the ubiquitin-proteasome pathway for efficient CoV replication (*Wong et al., 2015*; *Raaben et al., 2010a*). Indeed, this finding may provide a link to the described coronavirus RTC-encoded de-ubiquitination activity residing in nsp3 that has been implicated in innate immune evasion (*Barretto et al., 2005*; *Lindner et al., 2005*; *Bailey-Elkin et al., 2014*).

Most interestingly, this custom siRNA screen identified a crucial role of the host protein synthesis apparatus that was associated with the MHV RTC as indicated by the proximity-dependent proteomic screen (*Figure 3a*, *Figure 3c*). Silencing of ribosomal proteins Rpl13a and Rls24d1 and several subunits of the eIF3 complex resulted in greatly reduced MHV replication and scored with highest significance in the siRNA screen, suggesting that proximity of the host cell translation machinery to the viral RTC likely has functional importance for coronavirus replication (*Figure 4b*).

## Active translation near sites of viral mRNA synthesis

Due to the striking dependence of MHV replication on a subset of RTC-proximal translation initiation factors, we extended these results in independent assays. For this, we selected all host factors assigned to the category 'translation' (*Figure 3a*) and assessed virus replication following siRNA-mediated silencing of each factor. Measurement of luciferase activity after MHV-Gluc infection confirmed initial findings obtained by screening the entire siRNA library of MHV RTC-proximal factors (*Figure 4c*). Specifically for Rpl13a, and eIFs 3i, 3 f, and 3e viral replication was reduced to levels

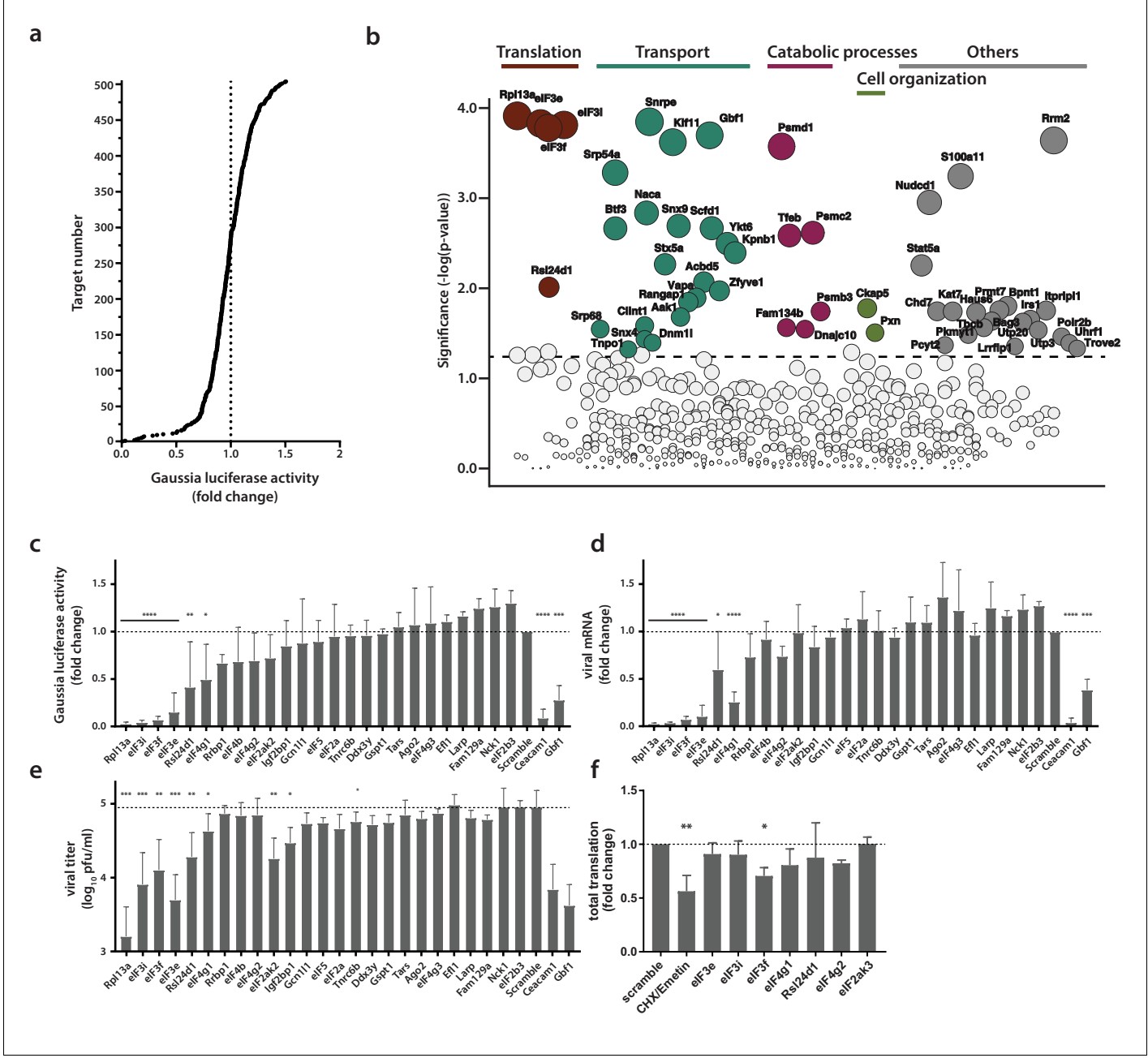

**Figure 4.** Identification of proviral factors within the coronavirus RTC microenvironment. (**a**) Impact of siRNA-silencing of RTC-proximal cellular proteins on viral replication. L929 fibroblasts were reverse-transfected with siRNAs (10 nM) for 48 hr before being infected with MHV-Gluc (MOI = 0.05, n = 4). Replication was assessed by virus-mediated Gaussia luciferase expression at 15 h.p.i. and was normalized to levels of viral replication in cells targeted by scrambled siRNA controls. Target proteins to the left of the dashed line represent RTC-proximal factors whose silencing decreased viral replication. (**b**) Bubble plot illustrating host proteins that significantly impact MHV replication. Bubble size is proportional to the level of viral replication impairment. Colors correspond to the functional categories highlighted in *Figure 3*. Light grey bubbles (below the dashed line) represent host proteins that did not significantly impact MHV replication (p-value > 0.05). (**c, d, e, f**) Silencing of RTC-proximal components of the cellular translation machinery. Upon 48 hr siRNA silencing of factors assigned to the category 'translation' (*Figure 3*), L929 fibroblasts were infected with MHV-Gluc (MOI = 0.05, n = 3). Luciferase activity (**c**), cell-associated viral RNA levels (**d**) and viral titers (**e**) were assessed at 12 h.p.i.. (**f**) Western blot quantification of total cellular translation following silencing of a subset of the host translation apparatus. Upon 48 hr siRNA-silencing, L929 fibroblasts were pulsed with 3 µM puromycin for 60 min. Control cells were treated, prior to puromycin incubation, with 355 µM cycloheximide and 208 µM Emetin for 30 min to block protein synthesis. Cell lysates were separated by SDS-PAGE and Western blots were probed using anti-puromycin antibodies to assess puromycin incorporation into polypeptides and normalized to actin levels. Error bars represent the mean ± standard deviation, where * is p ≤ 0.05, ** is p ≤ 0.005, *** is p ≤ 0.0005 and **** is p < 0.0001.

*Figure 4 continued on next page*

*Figure 4 continued*

DOI: https://doi.org/10.7554/eLife.42037.010

The following figure supplement is available for figure 4:

**Figure supplement 1.** (a) siRNA controls contained in each 96-well plate during siRNA-silencing of the RTC-proximal library.

DOI: https://doi.org/10.7554/eLife.42037.011

comparable to our controls Ceacam1a (MHV receptor) and Gbf1 (*Verheije et al., 2008*). Consistently, cell-associated viral mRNA levels (*Figure 4d*) and viral titers (*Figure 4e*) were reduced upon siRNA silencing of these factors. Although the silencing of a subset of host translation factors severely restricted MHV replication, effective knockdown of these factors (*Figure 4—figure supplement 1c*) did not affect cell viability (*Figure 4—figure supplement 1b*, *Figure 4—figure supplement 1d*) and only moderately affected host cell translation levels (*Figure 4f*, *Figure 4—figure supplement 1e*). This data demonstrates that the reduced viral replication observed after siRNA knockdown is not due to a general impairment of host translation. To confirm the knockdown of host translation factors on the protein level we employed antibodies that were available for eIF3e, eIF3f, and eIF3i, and as shown in *Figure 5*, murine L929 fibroblasts that were treated individually with four target-specific siRNAs displayed significantly reduced expression of eIF3e, eIF3f, and eIF3i proteins (*Figure 5a*, *Figure 5b*). Importantly, under conditions of eIF3e, eIF3f, and eIF3i knockdown, viral replication was also significantly restricted, confirming the importance of these translation initiation factors for MHV replication (*Figure 5c*).

Subsequently, we aimed to visualize the localization of active translation during virus infection by puromycin incorporation into nascent polypeptides on immobilized ribosomes (ribopuromycylation) followed by fluorescence imaging using antibodies directed against puromycin (*David et al., 2012*). In non-infected L929 cells, ribopuromycylation resulted in an expected diffuse, mainly cytosolic, staining pattern interspersed with punctate structures indicative of translation localized to dedicated subcellular cytosolic locations (*Figure 6*). In striking contrast, MHV-infected L929 cells displayed a pronounced enrichment of actively translating ribosomes near the viral RTC as indicated by the strong overlap between the viral replicase and the ribopuromycylation stain. Interestingly, active translation in vicinity of the RTC was strongest during the early phase of infection at 6 h.p.i., and was observed until 8 h.p.i., before gradually decreasing as the infection advanced along with the appearance of typical syncytia formation indicative of cytopathic effect (CPE).

Remarkably, we observed a similar phenotype in Huh7 cells infected with human coronaviruses, such as HCoV-229E or the highly pathogenic MERS-CoV (*Figure 7*). The HCoV-229E RTC, which was detected with an antiserum directed against nsp8, appeared as small and dispersed perinuclear puncta during early infection and eventually converged into larger perinuclear structures later in infection. Consistent with findings obtained for MHV, we observed a striking co-localization of the HCoV-229E RTC with sites of active translation during the early phase of the infection (*Figure 7*, *Figure 7—figure supplement 1*). The co-localization gradually decreased as the infection reached the late phase with upcoming signs of CPE. Finally, we further demonstrated that active translation is localized to the site of MERS-CoV RNA synthesis as dsRNA puncta highly overlapped with the ribopuromycylation stain in MERS-CoV-infected Huh7 cells (*Figure 7*). Collectively, these results not only confirm the spatial link between individual components of the host cell translation machinery and coronavirus replication compartments as identified by proximity-dependent biotinylation using MHV-BirA$_{R118G}$-nsp2, but they also demonstrate that active translation is taking place in close proximity to the viral RTC.

## Discussion

In this study, we made use of a recently developed system based on proximity-dependent biotinylation of host factors in living cells (*Roux et al., 2012*). By engineering a promiscuous biotin ligase (BirA$_{R118G}$) as an integral component of the coronavirus replication complex, we provide a novel approach to define the molecular mircoenvironment of viral replication complexes that is applicable to many other RNA and DNA viruses.

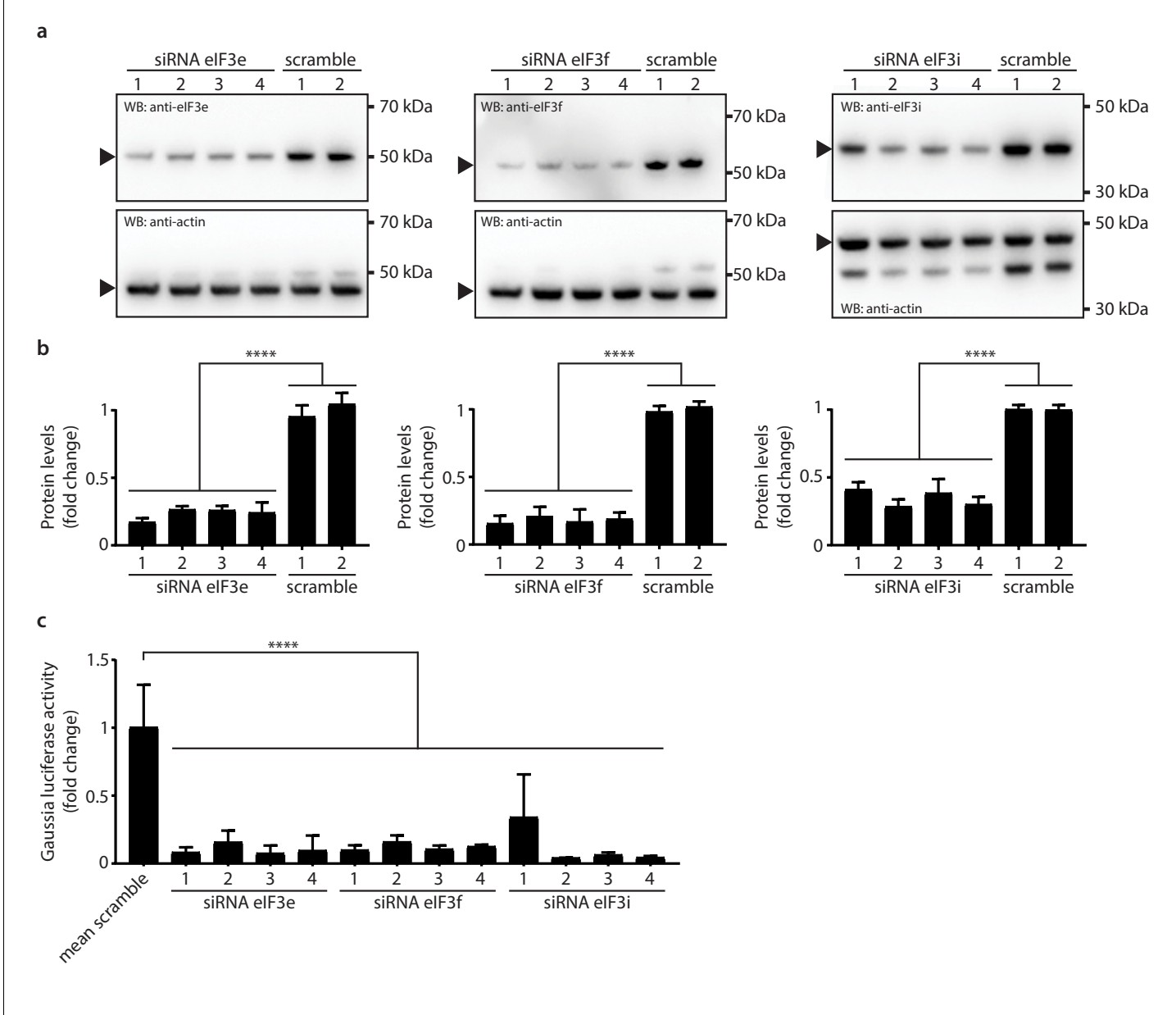

**Figure 5.** Knockdown of eIF3e, eIF3f and eIF3i impact MHV replication. (**a**) siRNA silencing and western blot assessment of eIF3e, eIF3f and eIF3i protein levels. 10 nM individual target-specific siRNA were transfected in L929 for 48 hr. Lysates were separated by SDS-PAGE and western blots were probed using antibodies directed against eIF3e, eIF3f and eIF3i. The same membranes were subsequently washed, and actin, which was used to normalize protein levels, was detected using a conjugated antibody. Arrows indicate proteins bands of interest. (**b**) protein quantification of eIF3e, eIF3f and eIF3i upon siRNA-mediated knockdown using single target-specific siRNAs (n = 3). Error bars represent the mean ± standard deviation, where **** is p < 0.0001. (**c**) L929 fibroblasts were infected with MHV-Gluc (MOI = 0.05, n = 3) for 12 hr upon a 48 hr knockdown of eIF3e, eIF3f and eIF3i using single target-specific siRNAs. Luciferase counts reflecting viral replication were normalized to scrambled non-targeting siRNA controls. Error bars represent the mean ± standard deviation, where **** is p < 0.0001.
DOI: https://doi.org/10.7554/eLife.42037.012

We show that nsp2 fusion proteins encoded by recombinant MHV-APEX2-nsp2 and MHV-Bir-A_{R118G}-nsp2, are indeed part of the RTC and localize to characteristic coronavirus replicative structures. On the ultrastructural level, APEX2-catalyzed DAB polymer depositions were detected at DMVs and CMs, and we observed co-localization of BirA_{R118G} with established coronavirus RTC markers, such as nsp2/3 and nsp8, by indirect immunofluorescence microscopy. Notably, in MHV-

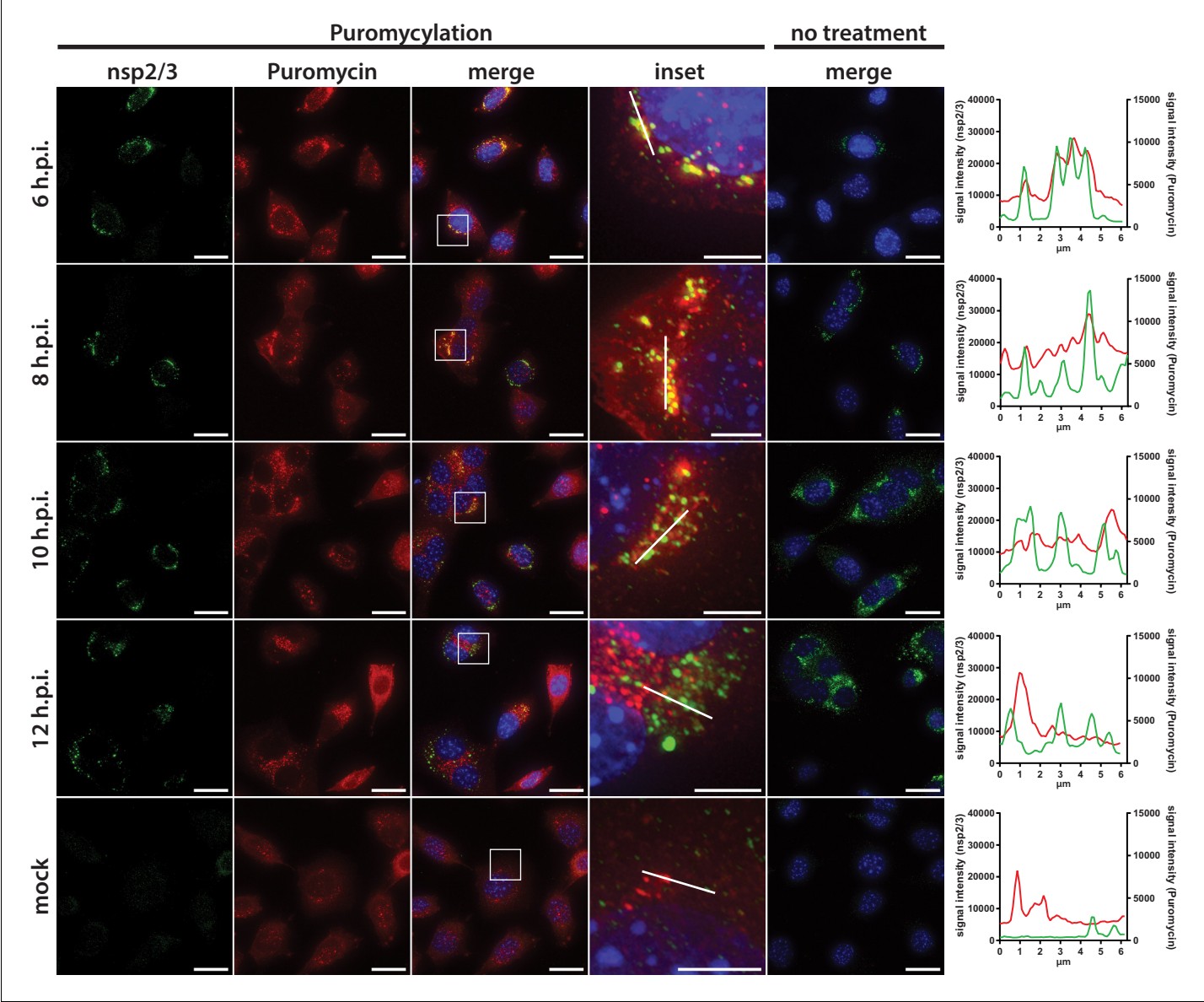

**Figure 6.** Active translation near sites of MHV mRNA synthesis. Visualization of active translation in MHV-infected L929 fibroblasts. Cells infected with MHV-A59 (MOI = 1) or non-infected cells were cultured for 6, 8, 10 and 12 hr and pulsed with cycloheximide, emetine and puromycin for 5 min to label translating ribosomes. All cells, including non-treated control infections, were subjected to a coextraction/fixation procedure to remove free puromycin. Cells were labeled using anti-nsp2/3 antiserum and anti-puromycin antibodies. Nuclei are counterstained with DAPI. Z-projection of deconvolved z-stacks acquired with a DeltaVision Elite High-Resolution imaging system are shown. Note the gradual decrease of overlap between the viral replication and actively translating ribosomes highlighted in the intensity profiles. Scale bar: 20 µm; insets 5 µm.
DOI: https://doi.org/10.7554/eLife.42037.013

BirA$_{R118G}$-nsp2-infected cells the detection of biotinylated coronavirus replicase gene products nsp2-10, nsp12-16, and the nucleocapsid protein by mass spectrometry demonstrates that these proteins are in close proximity during infection. This extends previous immunofluorescence and electron microscopic studies that were limited by the availability of nsp-specific antibodies and could only show localization of individual nsps to coronavirus replicative structures (*Knoops et al., 2008*; *Ulasli et al., 2010*; *Schiller et al., 1998*; *Hagemeijer et al., 2010*; *Graham et al., 2005*). Moreover, the close proximity of BirA$_{R118G}$-nsp2 to MHV replicative enzymes, such as the RNA-dependent RNA polymerase (nsp12), the NTPase/helicase (nsp13), the 5'-cap methyltransferases (nsp14, nsp16), the proof-reading exonuclease (nsp14), in MHV-BirA$_{R118G}$-nsp2-infected cells further suggests close

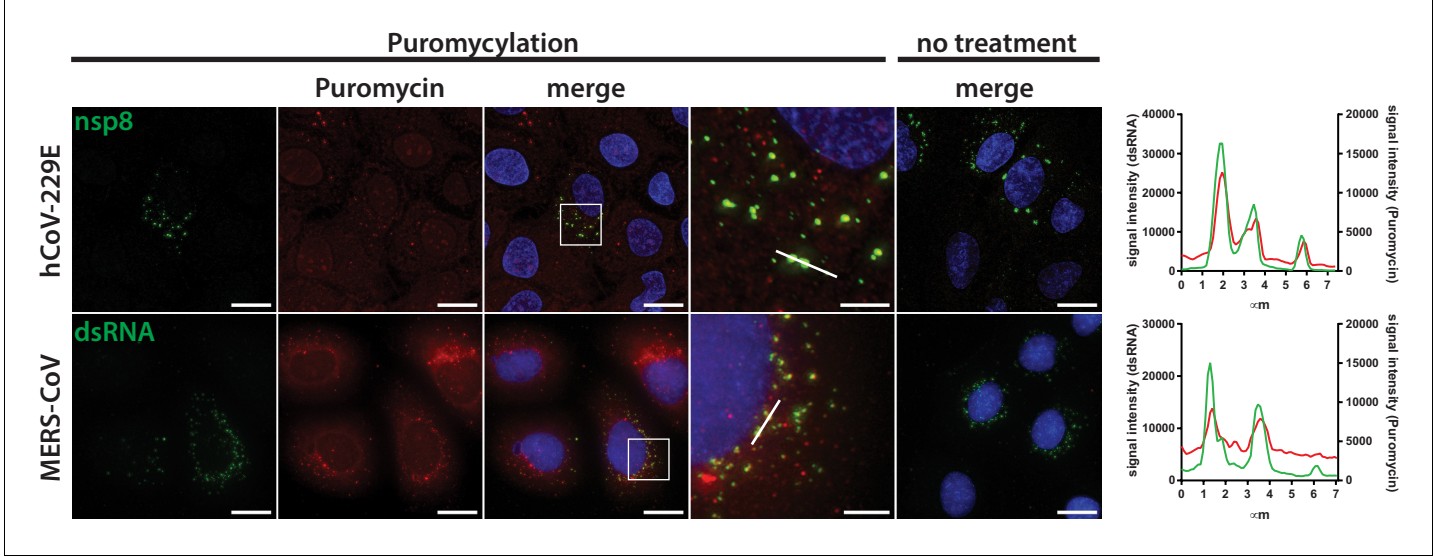

**Figure 7.** Active translation near sites of HCoV-229E and MERS-CoV mRNA synthesis. Visualization of active translation during HCoV-229E and MERS-CoV infections. Huh7 cells were infected with HCoV-229E and MERS-CoV (MOI = 1) for 12 hr and 6 hr, respectively. Cells were pulsed with cycloheximide, emetine and puromycin for 5 min to label translating ribosomes and subjected to a coextraction/fixation procedure to remove free puromycin. Non-infected and/or non-pulsed cells were used as control. Cells were labelled using anti-nsp8 (HCoV-229E) or dsRNA (MERS-CoV) and anti-puromycin antibodies. Nuclei are counterstained with DAPI. Z-projection of deconvolved z-stacks acquired with a DeltaVision Elite High-Resolution imaging system are shown. Intensity profiles in magnified regions are shown. Scale bar: 20 µm; insets 5 µm.
DOI: https://doi.org/10.7554/eLife.42037.014

The following figure supplement is available for figure 7:

**Figure supplement 1.** Visualization of active translation during HCoV-229E infections.
DOI: https://doi.org/10.7554/eLife.42037.015

proximity of nsp2 to the site of viral RNA synthesis. We thus propose that nsp2-16 and the nucleo-capsid protein collectively constitute a functional coronavirus replication and transcription complex in infected cells.

The analysis of the host proteome enriched at MHV replication sites revealed a comprehensive list of host proteins that constitute the coronavirus RTC microenvironment. This included several individual factors and host cell pathways, especially transport mechanisms involving vesicle-mediated trafficking, which have been shown to assist coronavirus replication. Indeed, previous findings have reported the importance of the early secretory pathway, as well as key proteins for these processes such as Gbf1 and Arf1, for efficient coronavirus replication (*Verheije et al., 2008*; *de Wilde et al., 2015*; *Oostra et al., 2007*; *Knoops et al., 2010*; *Vogels et al., 2011*; *Hsu et al., 2010*). Other markers such as Sec61α have also been detected in proximity of viral RTCs in SARS-CoV-infected cells (*Knoops et al., 2010*). Correspondingly, the implications of proteins involved in catabolic processes such as autophagy have also been linked to coronavirus replication and other positive-strand RNA viruses (*Reggiori et al., 2010*; *Sharma et al., 2014*; *Monastyrska et al., 2013*). Of note, the ubiquitin-proteasome system has been studied in more details during MHV infection and has also been highlighted in a genetic screen using infectious bronchitis coronavirus (*Wong et al., 2015*; *Raaben et al., 2010a*; *Raaben et al., 2010b*).

Notably, numerous coronavirus RTC-proximal host proteins and pathways also have documented roles in the life cycle of other, more intensively studied, positive-stranded RNA viruses. Recent genome-wide CRISPR screens identified proteins involved in biosynthesis of membrane and secretory proteins, as well as in the ERAD pathway, as required for flavivirus replication (*Marceau et al., 2016*; *Zhang et al., 2016*), suggesting considerable commonalities and conserved virus-host interactions at the replication complexes of a broad range of RNA viruses (*Marceau et al., 2016*; *Zhang et al., 2016*; *Hsu et al., 2010*; *Randall et al., 2007*).

Importantly, our list of RTC-proximal proteins by far exceeds the number of host cell proteins currently known to interact with viral replication complexes and the vast majority of MHV RTC-proximal

proteins have not been described before. These likely include proteins with defined temporal roles during particular phases of the viral life cycle and proteins that did not yet attract our attention in previous screens because of functional redundancies. We therefore expect that this approach will find wide application in the field of virus-host interaction, target identification for virus inhibition, and provides a starting point to reveal similarities and differences between replication strategies of a broad range of viruses.

One novel finding that arose immediately from our RTC-proximity screen is the demonstration of a close spatial association of host cell translation with the coronavirus RTC. Indeed, the biotin ligase-based proteomic screen identified a number of translation initiation factors, most prominently several eIF3 subunits that were found to have functional importance for viral replication, and numerous ribosome- and translation-associated proteins within the coronavirus RTC microenvironment (*Figure 3*, *Figure 4*). These results are in line with a recent genome-wide siRNA screen where translation factors were suggested to play a role in the replication of avian infectious bronchitis coronavirus (IBV) (*Wong et al., 2015*). The implication of this finding has, to our knowledge, not been further investigated. In addition, we noted the presence of subunits of the signal recognition particle in proximity to the coronavirus RTC and their functional relevance for viral replication, which is indicative of an importance for the translation of membrane proteins. Notably, the coronavirus RTC is translated as two polyproteins that contain nsp3, 4 and 6 with multiple trans-membrane domains that are believed to anchor the RTC at ER-derived membranes (*Knoops et al., 2008*; *Oostra et al., 2007*). It is thus tempting to speculate that the coronavirus RTC is either attracting, or deliberately forming in proximity to, the ER-localized host translation machinery in order to facilitate replicase translation and insertion into ER membranes. This idea is also applicable to many other positive-stranded RNA viruses that express viral polyproteins with embedded trans-membrane domains to anchor the viral replication complex in host endomembranes. Recent experimental evidence for Dengue virus supports this hypothesis. By using cell fractionation and ribosomal profiling, it has been shown that translation of the Dengue virus (family *Flaviviridae*) genome is associated with the ER-associated translation machinery accompanied by ER-compartment-specific remodeling of translation (*Reid et al., 2018*). Moreover, several recent genome-wide CRISPR screens demonstrated the functional importance of proteins involved in biosynthesis of membrane and secretory proteins, further supporting a pivotal role of the ER-associated translation machinery for virus replication (*Zhang et al., 2016*).

Compartmentalization of cellular translation to sites of viral RNA synthesis has been described for dsRNA viruses of the orthoreovirus family, which replicate and assemble in distinct cytosolic inclusions known as viral factories to which the host translation machinery is recruited (*Desmet et al., 2014*). The data presented here indicate that coronaviruses have evolved a similar strategy by compartmentalizing and directing viral RNA synthesis to sites of ER-associated translation. Likewise, this strategy has a number of advantages. Coronaviruses would not require sophisticated transport mechanisms that direct viral mRNA to distantly located ribosomes. A close spatial association of viral RNA synthesis and translation during early post-entry events would rather allow for remodeling the ER-associated translation machinery to ensure translation of viral mRNA in a protected microenvironment. Viruses have evolved diverse mechanisms to facilitate translation of their mRNAs including highly diverse internal ribosomal entry sites, recruitment of translation-associated host factors to viral RNAs, and even transcript-specific translation (*Hashem et al., 2013*; *Lee et al., 2013*). Accordingly, by remodeling defined sites for viral mRNA translation, the repertoire and concentration of translation factors can be restricted to factors needed for translation of these viral mRNAs. A microenvironment that is tailored towards the translational needs of viral mRNAs in proximity to the viral replicase complex would also make virus replication tolerant to host- or virus-induced shut down of translation at distal sites within the cytosol (*Raaben et al., 2007*). Notably, host translational shut down is well known for coronaviruses (*Raaben et al., 2007*) and specifically nsp1 has been implicated to play a role in this context by mediating host mRNA degradation (*Narayanan et al., 2015*; *Kamitani et al., 2006*). Coronaviruses may thus have evolved a two-pronged strategy to ensure efficient translation of viral proteins by establishing viral RNA synthesis in close proximity to actively translating ribosomes and by employing nsp1-mediated host cell mRNA degradation at RTC-distal sites. The strategy to assemble the viral RTC in close proximity to translation would also favor the coupling between genome translation and replication that has been proposed for picornaviruses and other positive-strand RNA viruses (*de Groot et al., 1992*; *Novak and Kirkegaard, 1994*).

Finally, host cells are equipped with fine-tuned mechanisms of foreign RNA recognition (*Gebhardt et al., 2017*). As such, MDA5 has been identified as a key cytosolic pattern recognition receptor restricting coronavirus replication (*Züst et al., 2011*). Likewise, the nonsense-mediated RNA decay pathway targets mRNAs of different origin containing aberrant features for degradation and has been newly demonstrated to also target cytosolic coronavirus mRNAs (*Wada et al., 2018*; *Schweingruber et al., 2013*). Therefore, proximity of viral mRNA synthesis and translation in a confined microenvironment protected from cytosolic surveillance factors can also be considered a mechanism to evade these cytosolic mRNA decay mechanisms and innate immune sensors of viral RNA.

The novel finding of a close association of the host translation machinery with sites of viral RNA synthesis during coronavirus infection exemplifies the power of the MHV-BirA$_{R118G}$-nsp2–mediated labeling approach to identify RTC-proximal cellular processes that significantly contribute to viral replication. Indeed, the ability of BirA$_{R118G}$ to label viral and host factors independently of high affinity and prolonged molecular interactions enables the establishment of a comprehensive repertoire reflecting the history of protein association with the viral RTC, recorded during the entire course of infection. In future studies it will be important to provide an 'RTC-association map' with temporal resolution. Like we have seen for translation initiation factors in this study, association of host cell proteins with the viral RTC might not persist throughout the entire replication cycle but might be of importance only transiently or during specific phases of the replication cycle. Given its short labeling time, APEX2 indeed offers this possibility to dissect protein recruitment to the viral RTC in a time-resolved manner, that is to detect RTC-associated host proteins at specific time points post infection. This will ultimately result in a dynamic, high resolution molecular landscape of virus-host interactions at the RTC and provide an additional impetus to elucidate critical virus-host interactions that take place at the site of viral RNA synthesis. These interactions should be exploited in the development of novel strategies to combat virus infection, based on conserved mechanisms of interactions at replication complexes of a broad range of positive-stranded RNA viruses.

# Materials and methods

**Key resources table**

| Reagent type (species) or resource | Designation | Source or reference | Identifiers | Additional information |
|---|---|---|---|---|
| Gene (Escherichia coli) | BirA$_{R118G}$ | | PMCID: 3308701 | |
| Strain, strain background (mouse hepatitis virus) | MHV-A59 | | PMID: 15709029 | |
| Strain, strain background (mouse hepatitis virus) | MHV-Gluc | | PMID: 24874215 | |
| Strain, strain background (mouse hepatitis virus) | MHV-BirA$_{R118G}$-nsp2 | This study | | |
| Strain, strain background (human coronavirus) | HCoV-229E | | PMID: 19057873 | |
| Strain, strain background (middle east respiratory syndrome coronavirus) | MERS-CoV | | PMID: 23170002; 23078800 | |
| Cell line (Mus musculus) | L929 | Sigma | 85011425 | |

*Continued on next page*

*Continued*

| Reagent type (species) or resource | Designation | Source or reference | Identifiers | Additional information |
|---|---|---|---|---|
| Cell line (Mus musculus) | 17Cl1 | Gift from S.G. Sawicki | PMC422565 | |
| Cell line (Homo sapiens) | Huh7 | Gift from V. Lohmann | CVCL_0336 | |
| Cell line (african green monkey) | Vero B4 | Gift from M. Müller | CVCL_1912 | |
| Antibody | anti-dsRNA J2 (mouse monoclonal IgG2a, kappa chain) | English and Scientific Consulting | Product No: 10010500 | 1:200 |
| Antibody | anti-myc (mouse monoclonal) | Cell signalling | 2276 | 1:8000 (IF); 1:1000 (WB) |
| Antibody | Anti-Nogo A + B (rabbit polyclonal) | Abcam | Product No : ab47085 | 1:200 |
| Antibody | Anti-EIF3E (rabbit polyclonal) | Sigma | HPA023973 | 1:100 (IF); 1:300 (WB) |
| Antibody | Anti-EIF3F (rabbit polyclonal) | Abcam | ab176853 | 1:3000 (WB) |
| Antibody | Anti-EIF3I (rabbit polyclonal) | Sigma | HPA029939 | 1:500 (WB) |
| Antibody | Anti-Puromycin (mouse monoclonal IgG2a,κ) | Merk Millipore | MABE343 | 1:10000 |
| Antibody | Anti-MHV nsp2/3 (rabbit polyclonal) | Gift from S. Baker | PMID: 9514967 | 1:200 |
| Antibody | Anti-MHV nsp8 (rabbit polyclonal) | Gift from S. Baker | PMID: 11907209 | 1:400 |
| Antibody | Anti-229E-nsp8 (rabbit polyclonal) | Gift from J Ziebuhr | PMID: 9847320 | 1:200 |
| Antibody | donkey anti-mouse 488 | Jackson ImmunoResearch | 715-545-150 | 1:400 |
| Antibody | donkey anti-rabbit 594 | Jackson ImmunoResearch | 711-585-152 | 1:400 |
| Antibody | donkey anti-rabbit 647 | Jackson ImmunoResearch | 711-605-152 | 1:400 |
| Antibody | donkey anti-rabbit HRP | Jackson ImmunoResearch | 711-035-152 | 1:10000 |
| Antibody | donkey anti-mouse HRP | Jackson ImmunoResearch | 715-035-151 | 1:5000 |
| Antibody | anti-actin HRP (mouse monoclonal) | Sigma | A3854 | 1:25000-1:50000 |
| Sequence-based reagent | On-Target Plus CherryPick siRNA Library | Horizon Discovery Ltd. | | |
| Commercial assay or kit | Pierce Gaussia Luciferase Glow Assay Kit | ThermoFisher Scientific | 16160 | |
| Commercial assay or kit | CytoTox 96 Non-Radioactive Cytotoxicity Assay | Promega | G1780 | |

*Continued on next page*

*Continued*

| Reagent type (species) or resource | Designation | Source or reference | Identifiers | Additional information |
|---|---|---|---|---|
| Commercial assay or kit | Viromer Green | Lipocalyx | VG-01LB-00 | |
| Commercial assay or kit | Dynabeads MyOne Streptavidin C1 | ThermoFisher Scientific | 65001 | |
| Chemical compound, drug | Puromycin | Sigma | P9620 | |
| Chemical compound, drug | Cycloheximide | Sigma | C7698 | |
| Chemical compound, drug | Emetin | Sigma | E2375 | |
| Chemical compound, drug | Biotin | Sigma | B4501 | |

## Cells

Murine L929 fibroblasts (ECACC 85011425) and murine 17Cl1 fibroblasts (gift from S.G. Sawicki) were cultured in MEM supplemented with 10% (v/v) heat-inactivated fetal bovine serum (FBS), 100 μg/ml streptomycin and 100 IU/ml penicillin (MEM+/+). Huh-7 hepatocarcinoma cells (gift from V. Lohnmann) and Vero B4 cells (kindly provided by M. Müller) were propagated in Dulbecco's Modified Eagle Medium-GlutaMAX supplemented with, 1 mM sodium pyruvate, 10% (v/v) heat-inactivated fetal bovine serum, 100 μg/ml streptomycin, 100 IU/ml penicillin and 1% (w/v) non-essential amino acids. 17Cl1 and Vero B4 are used routinely in our laboratory for the generation of virus stocks. L929 and Huh-7, which were used in this study's experiments, were newly purchased (L929) or were verified by a Multiplex human cell line authentication test in the Lohmann laboratory (Huh-7). All cell lines were regularly tested to check they were free of mycoplasma contamination using a commercially available system (LookOut Mycoplasma qPCR detection kit, Sigma).

## Viruses

Recombinant MHV strain A59 (WT), MHV-Gluc (*Lundin et al., 2014*), which expresses a *Gaussia* luciferase reporter replacing accessory gene 4 of MHV strain A59, and HCoV-229E were generated as previously described (*Coley et al., 2005*; *Eriksson et al., 2008*; *Thiel et al., 2001*). Viruses were propagated on 17Cl1 cells (MHV) and Huh-7 cells (HCoV-229E) and their sequence was confirmed by RT-PCR sequencing. MERS-CoV (*van Boheemen et al., 2012*; *Bermingham et al., 2012*) was propagated and titrated on Vero cells.

## Generation of recombinant MHV viruses

Recombinant MHV viruses were generated using a vaccinia virus-based system as described before (*Eriksson et al., 2008*). In short, a pGPT-1 plasmid encoding an *Escherichia coli* guanine phosphoribosyltransferase (GPT) flanked by MHV-A59 nt 447–950 and 1315–1774 was used for targeted homologous recombination with a vaccinia virus (VV) containing a full-length cDNA copy of the MHV-A59 genome (*Coley et al., 2005*). The resulting GPT-positive VV was further used for recombination with a plasmid containing the EGFP coding sequence flanked by MHV-A59 nt 477–956 and 951–1774 for the generation of MHV-GFP-nsp2, based on the strategy employed by Freeman et al. (*Freeman et al., 2014*). Alternatively, a plasmid containing the BirA$_{R118G}$ coding sequence (*Roux et al., 2012*) or the APEX2 coding sequence (*Lam et al., 2015*), with a N-terminal myc-tag or V5-tag, respectively, and a C-terminal (SGG)$_3$ flexible linker flanked by MHV-A59 nt 477–956 and 951–1774 was used for the generation of MHV-BirA$_{R118G}$-nsp2 and MHV-APEX2-nsp2. The resulting VV were used to generate full-length cDNA genomic fragments by restriction digestion of the VV backbone. Rescue of MHV-GFP-nsp2, MHV-BirA$_{R118G}$-nsp2 and MHV-APEX2-nsp2 was performed by

electroporation of capped in vitro transcribed recombinant genomes into a BHK-21-derived cell line stably expressing the nucleocapsid (N) protein layered on permissive 17Cl1 mouse fibroblasts. Recombinant MHV viruses were plaque-purified three times and purified viruses were passaged three times for stock preparations. All plasmid sequences, VV sequences and recombinant MHV sequences were confirmed by PCR or RT-PCR sequencing. Viruses were propagated on 17Cl1 cells and virus stocks were titrated by plaque assay on L929 cells.

## Viral replication assay

L929 cells were infected with MHV-A59, MHV-GFP-nsp2, MHV-BirA$_{R118G}$-nsp2 or MHV-APEX2-nsp2 in quadruplicate at an MOI = 1. Virus inoculum was removed 2 h.p.i., cells were washed with PBS and fresh medium was added. Viral supernatants were collected at the indicated time point and titrated by plaque assay on L929 cells. Titers reported are the averages of three independent experiments ± standard error of the mean (SEM).

## Immunofluorescence imaging

Biotinylation assays were carried out as described before with minor modifications (*Roux et al., 2013*). $10^6$ L929 cells grown on glass coverslips were infected with MHV-A59, MHV-BirA$_{R118G}$-nsp2 or MHV-APEX2-nsp2 at an MOI = 1, or non-infected in medium supplemented with 67 μM biotin (Sigma B4501). Cells were washed thrice with PBS at the indicated time points and fixed with 4% (v/v) neutral buffered formalin before being washed three additional times. Cells were permeabilized in PBS supplemented with 50 mM NH$_4$Cl, 0.1% (w/v) Saponin and 2% (w/v) BSA (CB) for 60 min and incubated 60 min with the indicated primary antibodies diluted in CB (polyclonal anti-MHV-nsp2/3 or nsp8 (gift from S Baker), 1:200 (*Schiller et al., 1998*; *Gosert et al., 2002*); anti-myc, 1:8000 Cell Signalling 2276). Cells were washed three times with CB and incubated for 60 min with donkey-derived, AlexaFluor488-conjugated anti-mouse IgG (H + L) and donkey-derived, AlexaFluor647-conjugated anti-rabbit IgG (H + L) (Jackson Immunoresearch). Cells were additionally labeled with streptavidin conjugated to AlexaFluor 594 (Molecular Probes) to detect biotinylated proteins. Coverslips were mounted on slides using ProLong Diamond Antifade mountant containing 4',6-diamidino-2-phenylindole (DAPI) (Thermo Fisher Scientific).

For indirect immunofluorescence detection of viral and host proteins, L929 cells were grown on glass coverslips in 24-well plates and infected with MHV-A59 or MHV-BirA$_{R118G}$-nsp2 (MOI = 1). At the indicated time point, cells were fixed with 4% (v/v) formalin and processed using primary monoclonal antibodies directed against dsRNA (J2 Mab, English Scientific and Consulting) or myc-tab (Cell signalling 2276) and polyclonal antibodies recognizing eIF3E (Sigma, HPA023973) or RTN4 (Nogo A + B, Abcam 47085) as well as secondary donkey-derived, AlexaFluor488-conjugated anti-mouse and AlexaFluor647-conjugated anti-rabbit IgG (H + L), as described above.

For proximity ligation assays, L929 cells were seeded in 24-well plates on glass coverslips and infected with MHV-A59 or MHV-BirA$_{R118G}$-nsp2 (MOI = 1). At the indicated time point, cells were washed with PBS, fixed with 4% (v/v) formalin and permeabilized with 0.1% (v/v) Triton X-100. Proximity ligation was performed as recommended by the manufacturer (Duolink In Situ detection reagents Red, Sigma) using monoclonal antibodies directed against dsRNA (J2, English and Scientific Consulting) or myc-tag (Cell Signaling 2276) and polyclonal antibodies recognizing eIF3E (Sigma, HPA023973) or RTN4 (Nogo A + B, Abcam 47085). Coverslips were mounted using Duolink In Situ Mounting Media with DAPI (Sigma).

All samples were imaged by acquiring 0.2 μm stacks over 10 μm using a DeltaVision Elite High-Resolution imaging system (GE Healthcare Life Sciences) equipped with a 60x or 100x oil immersion objective (1.4 NA). Images were deconvolved using the integrated softWoRx software and processed using Fiji (ImageJ). Brightness and contrast were adjusted identically for each condition and their corresponding control. Figures were assembled using the FigureJ plugin (*Mutterer and Zinck, 2013*).

## Biotinylation assay – western blot – mass spectrometry

L929 cells were infected with MHV-A59 or MHV-BirA$_{R118G}$-nsp2, and for comparison MHV$_{H277A}$ and MHV$_{H227A}$-BirA$_{R118G}$-nsp2, at an MOI = 1 in medium supplemented with 67 μM biotin (Sigma B4501). At 15 h.p.i., cells were washed three times with PBS and lysed in ice-cold buffer containing

50 mM TRIS-Cl pH 7.4, 500 mM NaCl, 0.2% (w/v) SDS, 1 mM DTT and 1x protease inhibitor (cOmplete Mini, Roche). Cells were scraped off the flask and transferred to tubes. Cells were kept on ice until the end of the procedure. Triton X-100 was added to each sample to a final concentration of 2%. Samples were sonicated for two rounds of 20 pulses with a Branson Sonifier 250 (30% constant, 30% power). Equal volumes of 50 mM TRIS-Cl were added to each sample and samples were centrifuged at 4°C for 10 min at 18,000 x $g$. Supernatants were incubated with magnetic beads on a rotator at 4°C overnight (800 µl Dynabeads per sample, MyOne Streptavidin C1, Life Technologies) that were previously washed with lysis buffer diluted 1:1 with 50 mM TRIS-Cl. Beads were washed twice with buffer 1 (2% (w/v) SDS), once with buffer 2 (0.1% (w/v) deoxycholic acid, 1% (v/v) Triton X-100, 1 mM EDTA, 500 mM NaCl, 50 mM HEPES pH 7.5), once with buffer 3 (0.5% w/v deoxycholic acid, 0.5% NP40, 1 mM EDTA, 250 mM LiCl, 10 mM TRIS-Cl pH 7.4) and once with 50 mM TRIS-Cl pH 7.4. Proteins were eluted from beads by the addition of 0.5 mM biotin and Laemmli SDS-sample buffer and heating at 95°C for 10 min.

For SDS-PAGE and western blot analysis, cells were cultured in six-well plates and lysates were prepared and affinity purified as described above. Proteins were separated on 10% (w/v) SDS-polyacrylamide gels (Bio-Rad), and proteins were electroblotted on nitrocellulose membranes (Amersham Biosciences, GE Healthcare) in a Mini Trans-Blot cell (Bio-Rad). Membranes were incubated in a protein-free blocking buffer (Advansta) and biotinylated proteins were probed by incubation with horseradish peroxidase-conjugated Streptavidin (Dako). Proteins were visualized using Western-Bright enhanced chemiluminescence horseradish peroxidase substrate (Advansta) according to the manufacturer's protocol.

For mass spectrometry analysis, lysates and affinity purification were performed as described above from $4*10^7$ cells cultured in 150 $cm^2$ tissue culture flasks. Proteins were separated 1 cm into a 10% (w/v) SDS-polyacrylamide gel. A Coomassie stain was performed and 4 × 2 mm bands were cut with a scalpel. Proteins on gel samples were reduced, alkylated and digested with Trypsin (*Gunasekera et al., 2012*). Digests were loaded onto a pre-column (C18 PepMap 100, 5 µm, 100 A, 300 µm i.d. x 5 mm length) at a flow rate of 20 µL/min with solvent C (0.05% TFA in water/acetonitrile 98:2). After loading, peptides were eluted in back flush mode onto the analytical Nano-column (C18, 3 µm, 100 Å, 75 µm x 150 mm, Nikkyo Technos C. Ltd., Japan) using an acetonitrile gradient of 5% to 40% solvent B (0.1% (v/v) formic acid in water/acetonitrile 4,9:95) in 40 min at a flow rate of 400 nL/min. The column effluent was directly coupled to a Fusion LUMOS mass spectrometer (Thermo Fischer, Bremen; Germany) via a nano-spray ESI source. Data acquisition was made in data-dependent mode with precursor ion scans recorded in the orbitrap with resolution of 120'000 (at m/z = 250) parallel to top speed fragment spectra of the most intense precursor ions in the Linear trap for a cycle time of 3 s maximum. Spectra interpretation was performed with Easyprot on a local, server run under Ubuntu against a forward + reverse *Mus musculus* 2016_04) and MHV 2016_07) database, using fixed modifications of carboamidomethylated on Cysteine, and variable modification of oxidation on Methionine, biotinylation on Lysine and on protein N-term, and deamidation of Glutamine and Asparagine. Parent and fragment mass tolerances were set to 10 ppm and 0.4 Da, respectively. Matches on the reversed sequence database were used to set a Z-score threshold, where 1% false discoveries (FDR) on the peptide spectrum match level had to be expected. Protein identifications were only accepted, when two unique peptides fulfilling the 1% FDR criterion were identified. MS identification of biotinylated proteins was performed in three independent biological replicates. For label-free protein quantification, LC-MS/MS data was interpreted with MaxQuant (version 1.5.4.1) using the same protein sequence databases and search parameters as for EasyProt. Match between runs was activated, however samples from different treatments were given non-consecutive fraction numbers in order to avoid over-interpretation of data. The summed and median normalized top3 peptide intensities extracted from the evidence table as a surrogate of protein abundance (*Braga-Lagache et al., 2016*) and LFQ values were used for statistical testing. The protein groups were first cleared from all identifications, which did not have at least two valid LFQ values. Protein LFQ levels derived from MaxQuant were log-transformed. Missing values were imputed by assuming a normal distribution between sample replicates. A two-tailed t-test was used to determine significant differences in protein expression levels between sample groups and p-values were adjusted for multiple testing using the Benjamini-Hochberg (FDR) test. The mass spectrometry proteomics data have been deposited to the ProteomeXchange Consortium via the PRIDE partner repository with the dataset identifier PXD009975.

## Computational analysis

Database for Annotation, Visualization, and Integrated Discovery (DAVID) was used to perform GO enrichment analysis on the RTC-proximal cellular factors identified via mass spectrometry (*Huang et al., 2009a*; *Huang et al., 2009b*; *Ashburner et al., 2000*; *The Gene Ontology Consortium, 2017*). GO BP terms with a p-value < 0.05 were considered to be terms that were significantly enriched in the dataset. Additional analysis of significant GO terms was conducted using AmiGO and revealed that the top 32 GO BP terms (p-value < 0.005) were predominantly associated with five broad functional categories (cell-cell adhesion, transport, cell organization, translation, and catabolic processes) (*Carbon et al., 2009*). Alternatively, enrichment analysis was performed using SetRank (data not shown), a recently described algorithm that circumvents pitfalls of commonly used approaches and thereby reduces the amount of false-positive hits (*Simillion et al., 2017*) and the following databases were searched for significant gene sets: BIOCYC (*Krummenacker et al., 2005*), GO (*Ashburner et al., 2000*), ITFP (*Zheng et al., 2008*), KEGG (*Kanehisa et al., 2014*), PhosphoSitePlus (*Hornbeck et al., 2012*), REACTOME (*Croft et al., 2014*), and WikiPathways (*Kelder et al., 2012*). Both independent approaches lead to highly similar results and consistently complement results obtained upon GO Cellular Components analysis.

STRING functional protein association networks were generated using RTC-proximal host proteins found within each of the five broad functional categories. Default settings were used for active interaction sources and a high confidence interaction score (0.700) was used to maximize the strength of data support. The MCL clustering algorithm was applied to each STRING network using an inflation parameter of 3 (*Szklarczyk et al., 2017*; *Szklarczyk et al., 2015*).

## siRNA screen

A custom siRNA library targeting each individual RTC-proximal factor (On Target Plus, SMART pool, 96-well plate format, Dharmacon, GE Healthcare) was ordered. Additionally, a deconvolved library of 4 individual siRNAs was purchased for selected targets. 10 nM siRNA were reverse transfected into L929 cells ($8*10^3$ cells per well) using Viromer Green (Lipocalyx) according to the manufacturer's protocol. Cells were incubated 48 hr at 37°C 5% $CO_2$ and cell viability was assessed using the Cyto-Tox 96 Non-Radioactive Cytotoxicity Assay (Promega). Cells were infected with MHV-Gluc (MOI = 0.05, 1000 plaque forming units/well), washed with PBS 3 h.p.i. and incubated in MEM+/ +for additional 9 or 12 hr. Gaussia luciferase was measured from the supernatant using Pierce Gaussia Luciferase Glow Assay Kit (ThermoFisher Scientific). Experiments were carried out in four independent replicates and both cytotoxicity values and luciferase counts were normalized to the corresponding non-targeting scrambled control of each plate. A one-way ANOVA (Kruskal-Wallis test, uncorrected Dunn's test) was used to test the statistical significance of reduced viral replication (mean <95% as compared to scramble control, n = 216). The R package ggplot2 was used to create the bubble plot (*Figure 4b*).

## siRNA screen validation

L929 cells were transfected with 10 nM siRNA as described above. 48 hr post-transfection, cell viability was assessed using the CytoTox 96 Non-Radioactive Cytotoxicity Assay (Promega) and visually inspected by automated phase-contrast microscopy using an EVOS FL Auto 2 Imaging System equipped with a 4x air objective. Cells were infected with MHV-Gluc (MOI = 0.05), washed with PBS 3 h.p.i. and incubated for 9 additional hours. *Gaussia* luciferase activity, viral titers and cell viability were measured from the supernatant as described above. One-way ANOVAs (ordinary one-way ANOVA, uncorrected Fisher's LSD test) were used to test the statistical significance.

Total cellular RNA was isolated from cells using the NucleoMag RNA Kit (Machery Nagel, Switzerland) on a KingFisher Flex Purification System (Thermo Fisher Scientific, Switzerland) according to the manufacture's instructions. The QuantiTect Probe RT-PCR Kit (Qiagen, Switzerland) was used according to the manufactures instructions for measuring the cell associated viral RNA levels with primers and probe specific to the MHV genome fragment coding the nucleocapsid gene (*Supplementary file 4*). Primers and Probe for mouse Glyceraldehyde 3-phosphate dehydrogenase (GAPDH) where obtained from ThermoFisher Scientific (Mm03302249_g1, Catalog Number: 4331182). The MHV levels were normalized to GAPDH and shown as ΔΔCt over mock (ΔCt values calculated as Ct reference - Ct target). The QuantiTect SYBR Green RT-PCR Kit (Qiagen, Switzerland)

was used according to the manufactures instructions for measuring the expression levels of Rpl13a, eIF3E, eIF3I, eIF3F, eIF4G1, eIF4G2, eIF2ak3, Rsl24d1 and Tbp. All primer pairs where placed over an exon intron junction (*Supplementary file 4*). All expression levels are displayed as ΔΔCt over non-targeting siRNA (ΔCt values calculated as Ct target - Ct Tbp) (*Livak and Schmittgen, 2001*). One-way ANOVA (ordinary one-way ANOVA, uncorrected Fisher's LSD test) was used to test the statistical significance.

Western blots were performed after a 48 hr transfection of 10 nM individual siRNAs as described before. Cells were lysed in M-PER Mammalian Protein Extraction Reagent (ThermoFisher Scientific) supplemented with cOmplete Mini Protease Inhibitor Cocktail (Roche) and Laemmli SDS-sample buffer. Samples were loaded on 4–12% Bolt Bis-Tris gels and run in MES SDS buffer (Life Technologies). Proteins were blotted on a nitrocellulose membranes using a power blotter system and power blotter select transfer stacks (ThermoFisher Scientific). Membranes were blocked in 5% milk in PBS supplemented with 0.5% Tween20 (PBST) and incubated with primary antibodies (anti-eIF3E, HPA023973; anti-eIF3F, ab176853; anti-eIF3I, HPA029939) and secondary HRP-conjugated donkey anti-rabbit antibodies (Jackson ImmunoResearch) in 0.5% milk in PBST. Proteins were visualized using WesternBright enhanced chemiluminescence horseradish peroxidase substrate (Advansta) according to the manufacturer's protocol. Subsequently, membranes were washed extensively in PBST and probed using an HRP-conjugated anti-actin antibody (Sigma A3854).

## Total cellular translation

siRNA-based silencing was performed as described above. 48 hr post-transfection, control cells were incubated with 355 μM cycloheximide (Sigma) and 208 μM Emetin (Sigma) for 30 min to block protein synthesis. Cells were treated with 3 μM puromycin for 60 min followed by three PBS washes (*Shen et al., 2018*). Total cell lysates were prepared using M-PER mammalian protein extraction reagent (Thermo Scientific) supplemented with protease inhibitors (cOmplete Mini, Roche). Lysates were separated on a 10% (w/v) SDS-PAGE and electroblotted as described above. Western blots were probed using a monclonal AlexaFluor647-conjugated anti-puromycin antibody (clone 12D10, Merk Millipore) and a donkey-derived HRP-conjugated anti-mouse (Jackson immunoresearch 715-035-151). Actin was detected using a monoclonal HRP-conjugated anti-actin antibody (Sigma A3854) and used to normalize input.

## Ribopuromycylation assay

Ribopuromycylation of actively translating ribosomes was performed as described before (*David et al., 2012*). L929, Huh-7 cells were seeded on glass coverslips and infected with MHV-A59 (L929), HCoV-229E (Huh-7) and MERS-CoV (Huh-7) at MOI = 1. One hour after inoculation, cells were washed with PBS and incubated further for the indicated time. Cells were treated with 355 μM cycloheximide and 208 μM Emetin (Sigma) for 15 min at 37°C. Cells were further incubated in medium containing 355 μM cycloheximide, 208 μM Emetin and 182 μM puromycin (Sigma) for additional 5 min. Cells were washed twice in ice-cold PBS and fix on ice for 20 min in buffer containing 50 mM TRIS HCl, 5 mM MgCl$_2$, 25 mM KCl, 355 μM cycloheximide, 200 mM NaCl, 0.1% (v/v) TritonX-100, 3% formalin and protease inhibitors (cOmplete Mini, Roche). Cells were blocked for 30 min in CB, and immunostained as described above using polyclonal anti-MHV-nsp2/3 (gift from S. Baker), polyclonal anti-HCoV-229E-nsp8 (gift from J. Ziebuhr), or monoclonal anti-dsRNA (J2 MAB, English and Scientific Consulting) as primary antibodies to detect MHV and HCoV-229E replication complexes, respectively. Donkey-derived, AlexaFluor488-conjugated anti-mouse or anti-rabbit IgG (H + L) were used as secondary antibodies. Additionally, ribosome-bound puromycin was detected using a monoclonal AlexaFluor647-conjugated anti-puromycin antibody (clone 12D10, Merk Millipore). Slides were mounted, imaged and processed as described above.

## DAB staining and transmission electron microscopy

L929 fibroblasts were seeded in 24-well plates and infected with MHV-APEX2-nsp2, MHV-A59, or non-infected for 10 hr. 3,3-diaminobenzidine (DAB) stains were performed as described previously (*Martell et al., 2017*). Briefly, cells were fixed at 10 h.p.i. using warm 2% (v/v) glutaraldehyde in 100 mM sodium cacodylate, pH 7.4, supplemented with 2 mM calcium chloride (cacodylate buffer) and placed on ice for 60 min. The following incubations were performed on ice in ice-cold buffers unless

stated otherwise. Cells were washed 3x with sodium cacodylate buffer, quenched with 20 mM glycine in cacodylate buffer for 5 min. before three additional washes with cacodylate buffer. Cells were stained in cacodylate buffer containing 0.5 mg/ml DAB and 10 mM H2O2 for 20 min until DAB precipitates were visible by light microscopy. Cells were washed 3x with cacodylate buffer to stop the staining reaction. Processing of samples for transmission electron microscopy (TEM) was performed as described previously (*Schätz et al., 2013*). Briefly, cells were washed once with PBS prewarmed to 37°C and subsequently fixed with 2.5% (v/v) glutaraldehyde (Merck, Darmstadt, Germany) in 0.1 M cacodylate buffer (Merck, Hohenbrunn, Germany) pH 7.4 for 30 min at room temperature or overnight at 4°C. After three washes in cacodylate buffer for 10 min each, cells were post-fixed with 1% OsO4 (Chemie Brunschwig, Basel, Switzerland) in 0.1 M cacodylate buffer for 1 hr at 4°C and again washed three times with cacodylate buffer. Thereafter, cells were dehydrated in an ascending ethanol series (70%, 80%, 90%, 94%, 100% (v/v) for 20 min each) and embedded in Epon resin, a mixture of Epoxy embedding medium, dodecenyl succinic anhydride (DDSA) and methyl nadic anhydride (MNA) (Sigma Aldrich, Buchs, Switzerland). Ultrathin sections of 90 nm were then obtained with diamond knives (Diatome, Biel, Switzerland) on a Reichert-Jung Ultracut E (Leica, Heerbrugg, Switzerland) and collected on collodion-coated 200-mesh copper grids (Electron Microscopy Sciences, Hatfield, PA). Sections were double-stained with 0.5% (w/v) uranyl acetate for 30 min at 40°C (Sigma Aldrich, Steinheim, Germany) and 3% (w/v) lead citrate for 10 min at 20°C (Laurylab, Saint Fons, France) in an Ultrastain (Leica, Vienna, Austria) and examined with a Philips CM12 transmission electron microscope (FEI, Eindhoven, The Netherlands) at an acceleration voltage of 80 kV. Micrographs were captured with a Mega View III camera using the iTEM software (version 5.2; Olympus Soft Imaging Solutions GmbH, Münster, Germany).

## Acknowledgements

We thank Mark Denison, Susan Baker, and John Ziebuhr for sharing virus sequence information and antisera. We thank Sandra Huber and Kerry Woods for helpful discussions.

This work was supported by the Swiss National Science Foundation (SNF; grants # 310030_173085, and # CRSII3_160780 to VT). SP was supported by the European Commission's Horizon 2020 research and innovation program under the Marie Skłodowska-Curie grant agreement no. 748627.

## Additional information

### Funding

| Funder | Grant reference number | Author |
| --- | --- | --- |
| Schweizerischer Nationalfonds zur Förderung der Wissenschaftlichen Forschung | 173085 | Philip V'kovski Volker Thiel |
| European Commission | 748627 | Stephanie Pfaender Volker Thiel |
| Schweizerischer Nationalfonds zur Förderung der Wissenschaftlichen Forschung | 160780 | Jenna Kelly Nadine Ebert Volker Thiel |

The funders had no role in study design, data collection and interpretation, or the decision to submit the work for publication.

### Author contributions

Philip V'kovski, Conceptualization, Formal analysis, Supervision, Validation, Investigation, Visualization, Methodology, Writing—original draft, Writing—review and editing; Markus Gerber, Formal analysis, Investigation, Visualization, Methodology; Jenna Kelly, Data curation, Formal analysis, Validation, Investigation, Visualization, Methodology, Writing—review and editing; Stephanie Pfaender, Nadine Ebert, Sophie Braga Lagache, Cedric Simillion, Jasmine Portmann, Hanspeter Stalder, Véronique Gaschen, Investigation, Methodology; Rémy Bruggmann, Supervision, Investigation,

Methodology; Michael H Stoffel, Supervision, Methodology, Writing—review and editing; Manfred Heller, Supervision, Investigation, Methodology, Writing—review and editing; Ronald Dijkman, Investigation, Methodology, Writing—review and editing; Volker Thiel, Conceptualization, Supervision, Funding acquisition, Methodology, Writing—original draft, Project administration, Writing—review and editing

## Author ORCIDs
Philip V'kovski (iD) http://orcid.org/0000-0002-8366-1220
Stephanie Pfaender (iD) http://orcid.org/0000-0002-3957-5448
Rémy Bruggmann (iD) http://orcid.org/0000-0003-4733-7922
Michael H Stoffel (iD) https://orcid.org/0000-0002-4699-5125
Manfred Heller (iD) http://orcid.org/0000-0002-6364-7325
Ronald Dijkman (iD) http://orcid.org/0000-0003-0320-2743
Volker Thiel (iD) http://orcid.org/0000-0002-5783-0887

## Decision letter and Author response
Decision letter https://doi.org/10.7554/eLife.42037.024
Author response https://doi.org/10.7554/eLife.42037.025

## Additional files

### Supplementary files
• Supplementary file 1. Mass spectrometry data.
DOI: https://doi.org/10.7554/eLife.42037.016

• Supplementary file 2. Gene Ontology enrichment analysis and STRING functional protein association association network analysis.
DOI: https://doi.org/10.7554/eLife.42037.017

• Supplementary file 3. siRNA-mediated knockdown screen.
DOI: https://doi.org/10.7554/eLife.42037.018

• Supplementary file 4. Primer and probes used in RT-qPCR.
DOI: https://doi.org/10.7554/eLife.42037.019

• Transparent reporting form
DOI: https://doi.org/10.7554/eLife.42037.020

### Data availability
The mass spectrometry proteomics data have been deposited to the ProteomeXchange Consortium via the PRIDE partner repository with the dataset identifier PXD009975. All other data generated or analysed during this study are included in the manuscript and supporting files.

The following dataset was generated:

| Author(s) | Year | Dataset title | Dataset URL | Database and Identifier |
|---|---|---|---|---|
| Manfred Heller | 2018 | Proximity biotinylation and LC-MSMS identification of the molecular microenvironment of coronavirus replication complexes | https://www.ebi.ac.uk/pride/archive/projects/PXD009975 | EBI PRIDE, PXD009975 |

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
