## [Decision Letter]

Thank you for submitting your article "Determination of host proteins composing the microenvironment of coronavirus replicase complexes by proximity-labeling" for consideration by *eLife*. Your article has been reviewed by two peer reviewers, and the evaluation has been overseen by a Reviewing Editor and Ivan Dikic as the Senior Editor. The reviewers have opted to remain anonymous.

The reviewers have discussed the reviews with one another and the Reviewing Editor has drafted this decision to help you prepare a revised submission.

Summary:

In this manuscript, V'kovski and colleagues described the use of proximity labeling in an attempt to define the complete proteome of replication complexes within the cytoplasm of coronavirus infected cells. For this study, the authors mainly focused on mouse hepatitis virus (MHV), but they also confirmed a subset of their findings with human coronavirus 229E and MERS-coronavirus. They used very well controlled experimental conditions at one time point during MHV infection of L929 cells in culture. The authors engineered a biotin ligase into the coding region of nsp2 and demonstrated that this virus, as well as a control GFP virus, had replication kinetics and growth properties similar to those of WT MHV. Five hundred proteins were identified, roughly 50 of which had proviral effects, as surmised from the decrease in viral RNA replication when the abundance of their mRNAs was reduced by siRNA treatment.

This is a well-crafted study that has generated a wealth of data regarding the composition of membrane bound replication complexes in coronavirus-infected mouse cells. It has the potential for uncovering the mechanistic details of how RNA replication complexes are assembled, how those complexes change during infection, and how viral non-structural proteins interact with a large array of host proteins to produce viral mRNAs and RNA replication intermediates. Given the already ambitious scope of the current study, all these mechanistic questions could not be addressed in this manuscript, but it is anticipated that data reported in this paper will be a springboard for many future studies. Here, the authors focused on translation factors and used a puromycin incorporation assay to show that active viral translation occurred near sites of viral RNA synthesis, providing evidence for the spatial coupling of viral translation and RNA synthesis functions and likely dependence of nascent protein synthesis for the proper topology of membrane association.

A small amount of additional protein work is needed to demonstrate the integrity of the BirA_R118G_-nsp2 fusion protein during infection and the reduction in ribosomal proteins following siRNA treatment.

Essential revisions:

1) Figure 1A and 1B. Given the propensity of viral proteins to be cleaved by viral and host proteinases, it is necessary to test whether all BirA_R118G_ remains fused to nsp2 in infected cells.

2) The data and their interpretation presented in the subsection “Active translation near sites of viral mRNA synthesis”, are problematic. It seems unlikely that knocking down canonical factors required for cap-dependent translation would dramatically impact viral translation but not host translation. The authors are using RT-PCR to verify knockdowns, but this would not account for the fact that some of these proteins may be very stable, especially if they are in translation initiation or elongation complexes. To deal with this possibility, it is essential for the authors to carry out Western blots following siRNA knockdowns of Rpl13a, eIF3i, eIF3f, eIF3e, and eIF4G1. The real readout here is protein levels, and this needs to be shown directly.

3) Subsection "Functional classification of the RTC-proximal host factors." This subsection contains a lengthy analysis of what emerged from the GO analysis, and how these observations could be connected with the literature. This is very interesting, but it would be good if the more speculative parts were moved to the Discussion. There, they could be combined with discussion of the 50 that emerged as being proviral following the siRNA screen. As it is, one is confused about what ideas were derived from the 500 candidates and what survived further investigation.

4) All discussion of the 500 proteins analyzed is based on the idea that they facilitate viral infection, or not. Surely there is a possibility that some of them are antiviral, and thus viral RNA replication will increase in their absence? Some acknowledgement of this possibility should be included.

---

## [Author Response]

Essential revisions:1) Figure 1A and 1B. Given the propensity of viral proteins to be cleaved by viral and host proteinases, it is necessary to test whether all BirA_R118G_ remains fused to nsp2 in infected cells.

The reviewers raised an important point. In order to assess whether BirA_R118G_ remains fused to nsp2 during infection, we performed a western blot analysis of mock-infected, MHV-WT and MHV-BirA_R118G_-nsp2-infected cells, using an anti-myc monoclonal antibody.

As shown in the new Figure 1—figure supplement 1, a specific band at the expected molecular weight of 102.2 kDa was detected specifically in MHV-BirA_R118G_-nsp2 lysates. Additionally, a band largely exceeding the size of the reference protein ladder likely represents polyprotein precursors. Importantly, no specific bands were detected at 36.7 kDa, which represents the expected molecular weight of myc-BirA_R118G_. Moreover, immunofluorescence analysis of MHV-BirA_R118G_-nsp2-infected cells further demonstrates the association of BirA_R118G_ with nsp2/3 and nsp8-labelled RTCs at all investigated time points (9, 12, 15 hours post infection) (Figure 1C and Figure 1—figure supplement 2, Figure 1—figure supplement 3). We included the immunofluorescence data from time points 9h and 15h post infection in the new supplementary Figure 1—figure supplement 2. Altogether, these data strongly suggest that BirA_R118G_ remains fused to nsp2 during the entire course of MHV-BirA_R118G_-nsp2 infection and is not cleaved by viral proteinases.

These results were included in the manuscript as two new supplementary figures (Figure 1—figure supplement 1 and Figure 1—figure supplement 2) and in the second paragraph of the subsection “Engineering the BirA_R118G_-biotin ligase into the MHV replicase transcriptase complex”.

2) The data and their interpretation presented in the subsection “Active translation near sites of viral mRNA synthesis”, are problematic. It seems unlikely that knocking down canonical factors required for cap-dependent translation would dramatically impact viral translation but not host translation. The authors are using RT-PCR to verify knockdowns, but this would not account for the fact that some of these proteins may be very stable, especially if they are in translation initiation or elongation complexes. To deal with this possibility, it is essential for the authors to carry out Western blots following siRNA knockdowns of Rpl13a, eIF3i, eIF3f, eIF3e, and eIF4G1. The real readout here is protein levels, and this needs to be shown directly.

We thank the reviewers for this comment, to which we entirely agree. In order to address this question, we ordered a deconvolved siRNA library containing target-specific, single siRNAs to silence eIF3e, eIF3f and eIF3i as well as eIF4G1 and Rpl13a. Next, we assessed protein levels by western blot after siRNA-mediated silencing of these factors (48h). For Rpl13a, we did not find suitable commercial antibodies, and western blots using both a polyclonal rabbit anti-eIF4G1 as well as a monoclonal rabbit anti-eIF4G1 (cell signalling 2858 and 8701) did not result in western blots of acceptable quality to allow quantification.

However, we could find suitable monoclonal antibodies for eIF3e, eIF3f and eIF3i, and we show in a new figure (Figure 5) that the knockdown of eIF3e, eIF3f and eIF3i, using 4 distinct siRNAs, resulted in a significant decrease of protein levels when normalized to an actin loading control. These data were performed in three independent siRNA knockdown followed by western blot experiments. Correspondingly, assessing viral replication using MHV-Gluc in the same knockdown conditions was consistent with previous results and confirmed that the knockdown of eIF3e, eIF3f and eIF3i significantly impairs viral replication.

These results are displayed in a new figure (Figure 5) and in the first paragraph of the subsection “Active translation near sites of viral mRNA synthesis”.

3) Subsection "Functional classification of the RTC-proximal host factors." This subsection contains a lengthy analysis of what emerged from the GO analysis, and how these observations could be connected with the literature. This is very interesting, but it would be good if the more speculative parts were moved to the Discussion. There, they could be combined with discussion of the 50 that emerged as being proviral following the siRNA screen. As it is, one is confused about what ideas were derived from the 500 candidates and what survived further investigation.

We thank the reviewers for this constructive remark. We adapted the subsection "Functional classification of the RTC-proximal host factors" accordingly and discussed parallels between published studies and our results in the Discussion section of the manuscript.

Statements referring to previously published work were removed from the Results section.

Accordingly, in the Discussion section, we discussed similarities between our results and other published studies using both coronaviruses and other positive-sense viruses. These changes are found in the third, fourth and fifth paragraph of the Discussion.

4) All discussion of the 500 proteins analyzed is based on the idea that they facilitate viral infection, or not. Surely there is a possibility that some of them are antiviral, and thus viral RNA replication will increase in their absence? Some acknowledgement of this possibility should be included.

We thank the reviewers for raising this point. Indeed, within the coronavirus RTC microenvironment, we expect to detect both proviral host factors that are required and support viral replication, as well as antiviral factors responsible for detecting viral replication intermediates. However since this study was performed in a L929 mouse fibroblast cell line, it is not surprising that the results are skewed towards highlighting proviral factors. To detect antiviral factors, other cell lines/types, e.g. primary macrophages, that are better equipped in eliciting antiviral responses, would be preferable.

Accordingly, we included the following sentence in the manuscript: “In contrast, we did not find antiviral factors that resulted in significant enhancement of viral replication upon siRNA knockdown. While this work was performed in a murine fibroblast cell line, the identification of antiviral proteins may be anticipated in a similar siRNA-mediated knockdown screen using primary target cells such as macrophages, that are better equipped in eliciting antiviral responses upon virus infection.”